# M3CoL: Harnessing Shared Relations via Multimodal Mixup Contrastive Learning for Multimodal Classification

**Raja Kumar**[‡1], **Raghav Singhal**[‡1], **Pranamya Kulkarni**[1], **Deval Mehta**[2], and **Kshitij Jadhav**[1]

[1]Indian Institute of Technology Bombay, Mumbai, India
[2]AIM for Health Lab, Department of Data Science & AI, Monash University, Australia

**Reviewed on OpenReview:** https://openreview.net/forum?id=NeQYi56MFj

## Abstract

Deep multimodal learning has shown remarkable success by leveraging contrastive learning to capture explicit one-to-one relations across modalities. However, real-world data often exhibits shared relations beyond simple pairwise associations. We propose **M3CoL**, a **M**ulti**m**odal **M**ixup **Co**ntrastive **L**earning approach to capture nuanced *shared relations* inherent in multimodal data. Our key contribution is a Mixup-based contrastive loss that learns robust representations by aligning mixed samples from one modality with their corresponding samples from other modalities thereby capturing shared relations between them. For multimodal classification tasks, we introduce a framework that integrates a fusion module with unimodal prediction modules for auxiliary supervision during training, complemented by our proposed Mixup-based contrastive loss. Through extensive experiments on diverse datasets (N24News, ROSMAP, BRCA, and Food-101), we demonstrate that **M3CoL** effectively captures shared multimodal relations and generalizes across domains. It outperforms state-of-the-art methods on N24News, ROSMAP, and BRCA, while achieving comparable performance on Food-101. Our work highlights the significance of learning shared relations for robust multimodal learning, opening up promising avenues for future research.

## 1 Introduction

The way we perceive the world is shaped by various modalities, such as language, vision, audio, and more. In the era of abundant and accessible multimodal data, it is increasingly crucial to equip artificial intelligence with multimodal capabilities (Baltrušaitis et al., 2018). At the heart of advancements in multimodal learning is contrastive learning, which maximizes similarity for positive pairs and minimizes it for negative pairs, making it practical for multimodal representation learning. CLIP (Radford et al., 2021) is a prominent example that employs contrastive learning to understand the direct link between paired modalities and seamlessly maps images and text into a shared space for cross-modal understanding, which can be later utilized for tasks such as retrieval and classification. However, traditional contrastive learning methods often overlook shared relationships between samples across different modalities, which can result in the learning of representations that are not fully optimized for capturing the underlying connections between diverse data modalities. These methods primarily focus on distinguishing between positive and negative pairs of samples, typically treating each instance as an independent entity. They tend to disregard the rich, shared relational information that could exist between samples within and across modalities. This limited focus can prevent the model from leveraging valuable contextual information, such as semantic similarities or complementary patterns, which can enhance robust representation learning. Consequently, this can lead to suboptimal performance in downstream tasks that require optimized shared representations, such as image-text alignment, cross-modal retrieval, or multimodal fusion tasks.

---

[‡]Equal Contributions. Author ordering determined by coin flip over Google Meet.
Our code is available at: https://github.com/RaghavSinghal10/M3CoL.

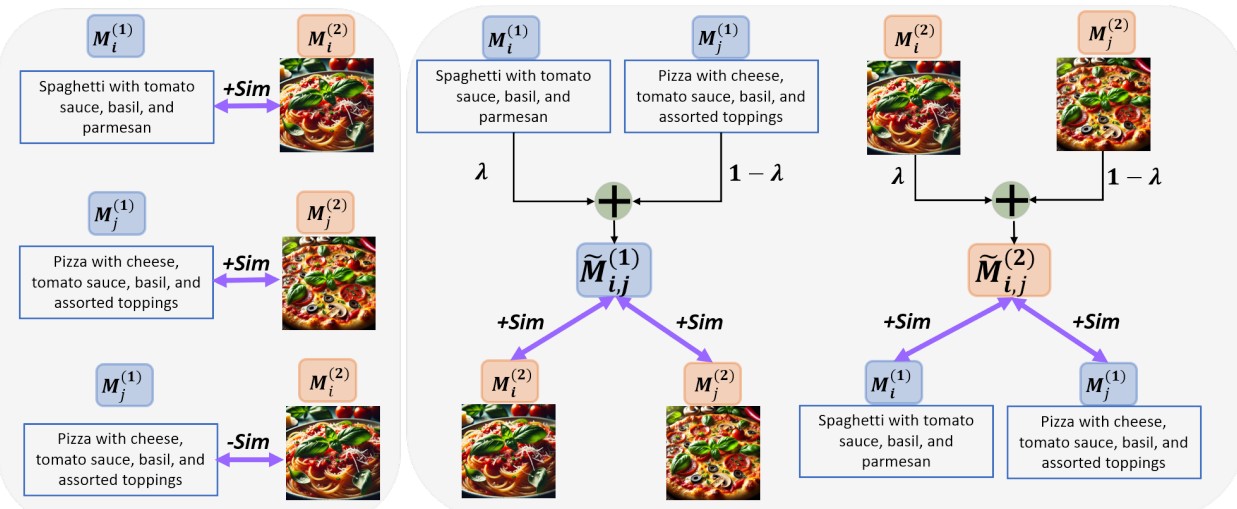

Figure 1: Comparison of traditional contrastive and our proposed M3Co loss. $\mathbf{M}_i^{(1)}$ and $\mathbf{M}_i^{(2)}$ denote representations of the $i$-th sample from modalities 1 and 2, respectively. Traditional contrastive loss (left panel) aligns corresponding sample representations across modalities. M3Co (right panel) mixes the $i$-th and $j$-th samples from modality 1 and enforces the representations of this mixture to align with the representations of the corresponding $i$-th and $j$-th samples from modality 2, and vice versa. For the text modality, we mix the text embeddings, while we mix the raw inputs for other modalities. Similarity (Sim) represents type of alignment enforced between the embeddings for all modalities.

While traditional contrastive learning methods treat paired modalities as positive samples and non-corresponding ones as negative, they often overlook shared relations between different samples. As shown in the left panel of Figure 1 (Left panel), classical contrastive learning approach assumes perfect one-to-one relations between modalities, which is rare in real-world data. For example, shared elements in images or text can relate even across separate samples, as illustrated by the elements like *"tomato sauce"* and *"basil"* in Figure 1. Our approach, illustrated in the right panel of Figure 1, goes beyond simple pairwise alignment by capturing shared relationships across mixed samples. By creating newer data points through convex combinations of data points our method more effectively models complex shared relationships, such as imperfect bijections (Liang et al., 2022a), enhancing multimodal classification performance.

Our approach builds upon the success of data augmentation techniques such as Mixup (Zhang et al., 2017) and their variants (Yun et al., 2019; Cubuk et al., 2019; Hendrycks et al., 2019), which have proven beneficial for enhancing learned feature spaces, improving both robustness and performance. Mixup trains models on synthetic data created through convex combinations of two datapoint-label pairs (Chapelle et al., 2000). These techniques are particularly valuable in low sample settings, as they help prevent overfitting and the learning of ineffective shortcuts (Chen et al., 2020; Robinson et al., 2021), common in contrastive learning. Building on the success of recent Mixup strategies (Shen et al., 2022; Thulasidasan et al., 2019; Verma et al., 2019) and MixCo (Kim et al., 2020), we introduce M3Co, a novel approach that significantly adapts and enhances contrastive learning principles to complex multimodal settings. M3Co modifies the CLIP loss to effectively handle multimodal scenarios, addressing the problem of instance discrimination, where models overly focus on distinguishing individual instances instead of capturing relationships between modalities. By leveraging convex combinations of data for contrastive learning, M3Co eliminates instance discrimination and enhances robust representation learning by capturing shared relations. These combinations serve as structured noise and treated as positive pairs with their corresponding samples from other modalities. Our experimental results demonstrate enhanced ability to capture shared relations enabling improvements in performance and generalization across a range of multimodal classification tasks.

Our key contributions are summarized as follows:

- We propose M3Co, a multimodal contrastive loss (Eq. 8) that utilizes mixed samples to effectively capture shared relationships across different modalities. By going beyond traditional pairwise alignment methods, M3Co makes representations more consistent with the complex, intertwined relationships usually observed in real-world data.

- We introduce a multimodal learning framework (Figure 2) consisting of unimodal prediction modules, a fusion module, and a novel Mixup-based contrastive loss. Our proposed method is modality-agnostic, allowing for flexible application across various types of data, and continuously updates the representations necessary for accurate and consistent predictions.

- We demonstrate the effectiveness of our methodology by evaluating it on four public multimodal classification benchmark datasets from different domains: two image-text datasets, N24News and Food-101, and two medical datasets, ROSMAP and BRCA (Table 1, 2, 3). Our approach outperforms baseline models, especially on smaller datasets.

## 2 Methodology

**Pipeline Overview:** Figure 2 depicts our framework, which comprises of three components: unimodal prediction modules, a fusion module, and a Mixup-based contrastive loss. We obtain latent representations (using learnable modality specific encoders $f^{(1)}$ and $f^{(2)}$) of individual modalities and fuse them (denoted by concatenation symbol '+') to generate a joint multimodal representation, which is optimized using a supervised objective (through classifier 3). The unimodal prediction modules provide additional supervision during training (via classifier 1 and 2). These strategies enable deeper integration of modalities and allow the models to compensate for the weaknesses of one modality with the strengths of another. The Mixup-based contrastive loss (denoted by $\mathcal{L}_{\mathrm{M3Co}}$) continuously updates the representations by capturing shared relations inherent in the multimodal data. This comprehensive approach enhances the understanding of multimodal data, improving accuracy and model robustness.

**Multimodal Mixup Contrastive Learning:** Given a batch of $N$ multimodal samples, let $\mathbf{x}_i^{(1)}$ and $\mathbf{x}_i^{(2)}$ denote the $i$-th samples for the first and second modalities, respectively. The modality encoders, $f^{(1)}$ and $f^{(2)}$, generate the corresponding embeddings $\mathbf{p}_i^{(1)}$ and $\mathbf{p}_i^{(2)}$:

$$\mathbf{p}_i^{(1)} = f^{(1)}(\mathbf{x}_i^{(1)}), \quad \mathbf{p}_i^{(2)} = f^{(2)}(\mathbf{x}_i^{(2)}) \tag{1}$$

We generate a mixture, $\tilde{\mathbf{x}}_{i,j}^{(1)}$, of the samples $\mathbf{x}_i^{(1)}$ and $\mathbf{x}_j^{(1)}$ by taking their convex combination. Similarly, we generate a mixture, $\tilde{\mathbf{x}}_{i,k}^{(2)}$, using the convex combination of the samples $\mathbf{x}_i^{(2)}$ and $\mathbf{x}_k^{(2)}$ (Eq. 2). In the case of text modality, instead of directly mixing the raw inputs, we mix the text embeddings (Guo et al., 2019). The mixing indices $j, k$ are drawn arbitrarily, without replacement, from $[1, N]$, for both the modalities. We mix both the modalities using a factor $\lambda \sim \text{Beta}(\alpha, \alpha)$. Based on the findings of (Zhang et al., 2017), which demonstrated enhanced performance for $\alpha$ values between 0.1 and 0.4, we chose $\alpha = 0.15$ after experimenting with several values in this range. The mixtures are fed through the respective encoders to obtain the embeddings: $\tilde{\mathbf{p}}_{i,j}^{(1)}$, and $\tilde{\mathbf{p}}_{i,k}^{(2)}$ (Eq. 3).

$$\tilde{\mathbf{x}}_{i,j}^{(1)} = \lambda_i \cdot \mathbf{x}_i^{(1)} + (1 - \lambda_i) \cdot \mathbf{x}_j^{(1)}, \quad \tilde{\mathbf{x}}_{i,k}^{(2)} = \lambda_i \cdot \mathbf{x}_i^{(2)} + (1 - \lambda_i) \cdot \mathbf{x}_k^{(2)} \tag{2}$$

$$\tilde{\mathbf{p}}_{i,j}^{(1)} = f^{(1)}(\tilde{\mathbf{x}}_{i,j}^{(1)}), \quad \tilde{\mathbf{p}}_{i,k}^{(2)} = f^{(2)}(\tilde{\mathbf{x}}_{i,k}^{(2)}) \tag{3}$$

The unidirectional contrastive loss (Sohn, 2016; Chen et al., 2020; Oord et al., 2018; Wu et al., 2018; Zhang et al., 2022) over $\mathbf{p}^{(2)}$ is conventionally defined as:

$$\mathcal{L}_{\text{sim-conv}}(\mathbf{p}^{(1)}, \mathbf{p}^{(2)}) = -\frac{1}{N} \sum_{i=1}^{N} \log \frac{\exp\left(\mathbf{p}_i^{(1)} \cdot \mathbf{p}_i^{(2)}/\tau\right)}{\sum_{j=1}^{N} \exp\left(\mathbf{p}_i^{(1)} \cdot \mathbf{p}_j^{(2)}/\tau\right)} \tag{4}$$

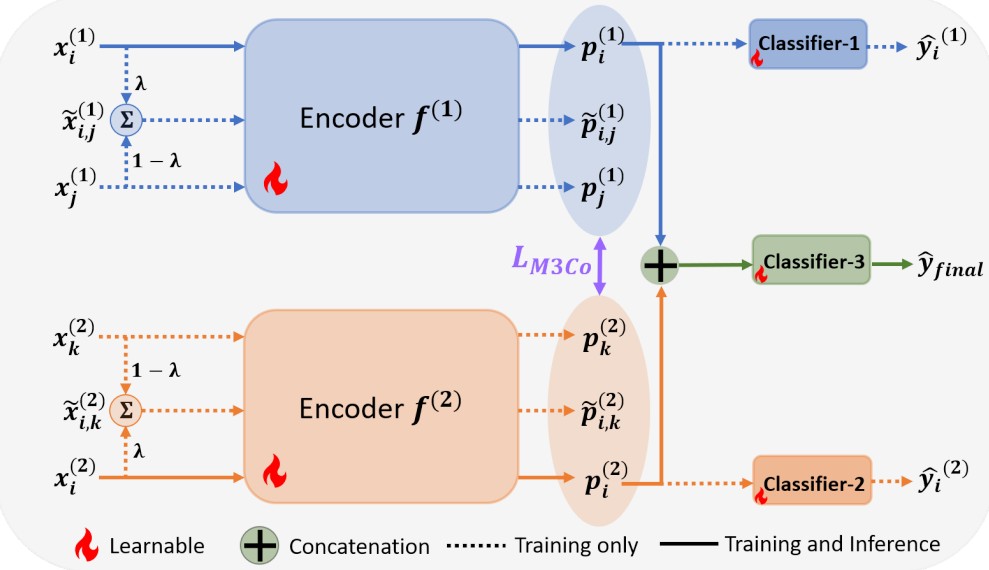

Figure 2: Architecture of our proposed M3CoL model. Samples from modality 1 ($\mathbf{x}_i^{(1)}$, $\mathbf{x}_j^{(1)}$) and modality 2 ($\mathbf{x}_i^{(2)}$, $\mathbf{x}_k^{(2)}$), along with their respective mixed data $\tilde{\mathbf{x}}_{i,j}^{(1)}$ and $\tilde{\mathbf{x}}_{i,k}^{(2)}$, are fed into encoders $f^{(1)}$ and $f^{(2)}$ to generate embeddings. Unimodal embeddings $\mathbf{p}_i^{(1)}$ and $\mathbf{p}_i^{(2)}$ are processed through classifier 1 and 2 to produce predictions $\hat{\mathbf{y}}_i^{(1)}$ and $\hat{\mathbf{y}}_i^{(2)}$ for training supervision only. The unimodal embeddings $\mathbf{p}_i^{(1)}$ and $\mathbf{p}_i^{(2)}$ are concatenated and processed through classifier 3 to yield $\hat{\mathbf{y}}_{\text{final}}$, utilized during training and inference. Additionally, unimodal embeddings $\mathbf{p}_i^{(1)}$, $\mathbf{p}_j^{(1)}$, $\mathbf{p}_i^{(2)}$, $\mathbf{p}_k^{(2)}$, and mixed embeddings $\tilde{\mathbf{p}}_{i,j}^{(1)}$ and $\tilde{\mathbf{p}}_{i,k}^{(2)}$ are utilized by our contrastive loss $\mathcal{L}_{\text{M3Co}}$ for shared alignment.

where $\cdot$ indicates dot product and $\tau$ is a temperature hyperparameter. While this formulation is needed for computing similarity among aligned samples from different modalities, our loss handles both aligned and non-aligned samples, as this enables to learn a better representation space. To achieve this, we define the unidirectional multimodal contrastive loss between $\mathbf{p}_i^{(1)}$ and $\mathbf{p}_m^{(2)}$ over $\mathbf{p}^{(2)}$ as:

$$\mathcal{L}_{\text{sim}}(\mathbf{p}_i^{(1)}, \mathbf{p}^{(2)}; m) = -\log \frac{\exp\left(\mathbf{p}_i^{(1)} \cdot \mathbf{p}_m^{(2)}/\tau\right)}{\sum_{j=1}^{N} \exp\left(\mathbf{p}_i^{(1)} \cdot \mathbf{p}_j^{(2)}/\tau\right)} \tag{5}$$

where $\mathbf{p}^{(1)}$ and $\mathbf{p}^{(2)}$ are $\mathcal{L}^2$ normalized, $\tau$ is a temperature hyperparameter, and $m$ is a sample index in $[1, N]$. Although the unidirectional multimodal contrastive loss (Eq. 5) can learn indirect relations, it is insufficient for learning shared semi-positive relations between modalities. Therefore, we introduce a Mixup-based contrastive loss to capture these relations that promotes generalized learning, as this process is more nuanced than simply discriminating positives from negatives. Now, we make our loss bidirectional to encourage improved alignment in the shared representation space and efficient use of training data (Radford et al., 2021; Oord et al., 2018; Sohn, 2016). We define this bidirectional Mixup contrastive loss M3Co for each modality (Eq. 6, 7) and the total M3Co loss (Eq. 8) as:

$$\mathcal{L}_{\text{M3Co}}^{(1)} = \frac{1}{N}\sum_{i=1}^{N}\left[\lambda_i \cdot \mathcal{L}_{\text{sim}}(\tilde{\mathbf{p}}_{i,j}^{(1)}, \mathbf{p}^{(2)}; i) + (1-\lambda_i) \cdot \mathcal{L}_{\text{sim}}(\tilde{\mathbf{p}}_{i,j}^{(1)}, \mathbf{p}^{(2)}; j)\right]$$

$$+ \frac{1}{N}\sum_{i=1}^{N}\left\{\lambda_i \cdot \mathcal{L}_{\text{sim}}(\mathbf{p}_i^{(2)}, \tilde{\mathbf{p}}^{(1)}; i) + (1-\lambda_i) \cdot \mathcal{L}_{\text{sim}}(\mathbf{p}_j^{(2)}, \tilde{\mathbf{p}}^{(1)}; i)\right\} \tag{6}$$

$$\mathcal{L}_{\text{M3Co}}^{(2)} = \frac{1}{N} \sum_{i=1}^{N} \left[ \lambda_i \cdot \mathcal{L}_{\text{sim}}(\tilde{\mathbf{p}}_{i,k}^{(2)}, \mathbf{p}^{(1)}; i) + (1 - \lambda_i) \cdot \mathcal{L}_{\text{sim}}(\tilde{\mathbf{p}}_{i,k}^{(2)}, \mathbf{p}^{(1)}; k) \right]$$

$$+ \frac{1}{N} \sum_{i=1}^{N} \left\{ \lambda_i \cdot \mathcal{L}_{\text{sim}}(\mathbf{p}_i^{(1)}, \tilde{\mathbf{p}}^{(2)}; i) + (1 - \lambda_i) \cdot \mathcal{L}_{\text{sim}}(\mathbf{p}_k^{(1)}, \tilde{\mathbf{p}}^{(2)}; i) \right\} \tag{7}$$

$$\mathcal{L}_{\text{M3Co}}^{(1,2)} = \frac{1}{2} \left( \mathcal{L}_{\text{M3Co}}^{(1)} + \mathcal{L}_{\text{M3Co}}^{(2)} \right) \tag{8}$$

where $\mathbf{p}^{(1)}$, $\tilde{\mathbf{p}}^{(1)}$, $\mathbf{p}^{(2)}$, and $\tilde{\mathbf{p}}^{(2)}$ are $\mathcal{L}^2$ normalized. Note that the parts of the loss functions in Eq. (6, 7) inside curly parenthesis make them bidirectional. Mixup-based methods enhance generalization by capturing clean patterns in the early training stages but can eventually overfit to noise if continued for larger number of epochs (Liu et al., 2023; Yu et al., 2021; Golatkar et al., 2019). To address this, we implement a schedule that transitions from the Mixup-based M3Co loss to a non-Mixup multimodal contrastive loss. We design this transition so that the non-Mixup loss retains the ability to learn shared or indirect relationships between modalities. By using a bidirectional SoftClip-based loss (Gao et al., 2024; Sohn, 2016; Chen et al., 2020), we relax the rigid one-to-one correspondence, allowing the model to capture many-to-many relations (Gao et al., 2024; 2022). The bidirectional **MultiSoftClip** loss for each modality (Eq. 9, 10) and its combination (Eq. 11) is:

$$\mathcal{L}_{\text{MultiSClip}}^{(1)} = \frac{1}{N} \sum_{i=1}^{N} \sum_{l=1}^{N} \left[ \frac{\exp\left(\mathbf{p}_i^{(1)} \cdot \mathbf{p}_l^{(1)}/\tau\right)}{\sum\limits_{t=1}^{N} \exp\left(\mathbf{p}_i^{(1)} \cdot \mathbf{p}_t^{(1)}/\tau\right)} \cdot \left( \mathcal{L}_{\text{sim}}(\mathbf{p}_i^{(2)}, \mathbf{p}^{(1)}; l) + \mathcal{L}_{\text{sim}}(\mathbf{p}_l^{(1)}, \mathbf{p}^{(2)}; i) \right) \right] \tag{9}$$

$$\mathcal{L}_{\text{MultiSClip}}^{(2)} = \frac{1}{N} \sum_{i=1}^{N} \sum_{l=1}^{N} \left[ \frac{\exp\left(\mathbf{p}_i^{(2)} \cdot \mathbf{p}_l^{(2)}/\tau\right)}{\sum\limits_{t=1}^{N} \exp\left(\mathbf{p}_i^{(2)} \cdot \mathbf{p}_t^{(2)}/\tau\right)} \cdot \left( \mathcal{L}_{\text{sim}}(\mathbf{p}_i^{(1)}, \mathbf{p}^{(2)}; l) + \mathcal{L}_{\text{sim}}(\mathbf{p}_l^{(2)}, \mathbf{p}^{(1)}; i) \right) \right] \tag{10}$$

$$\mathcal{L}_{\text{MultiSClip}}^{(1,2)} = \frac{1}{2} \left( \mathcal{L}_{\text{MultiSClip}}^{(1)} + \mathcal{L}_{\text{MultiSClip}}^{(2)} \right) \tag{11}$$

where $\mathbf{p}^{(1)}$ and $\mathbf{p}^{(2)}$ are $\mathcal{L}^2$ normalized. The M3Co and MultiSClip losses for $M$ modalities is:

$$\mathcal{L}_{\text{M3Co}} = \sum_{i=1}^{M} \sum_{j>i}^{M} \mathcal{L}_{\text{M3Co}}^{(i,j)} \tag{12}$$

$$\mathcal{L}_{\text{MultiSClip}} = \sum_{i=1}^{M} \sum_{j>i}^{M} \mathcal{L}_{\text{MultiSClip}}^{(i,j)} \tag{13}$$

**Unimodal Predictions and Fusion:** The encoders produce latent representations for each of the $M$ modalities, serving as inputs to individual classifiers that generate modality-specific predictions $\hat{\mathbf{y}}^{(m)}$. These representations are used for modality-specific supervision only during training. The unimodal prediction task involves minimizing the cross-entropy loss $\mathcal{L}_{\text{CE}}$ between these predictions and the corresponding ground truth labels ($\mathbf{y}$), for each modality. The unimodal cross-entropy loss is:

$$\mathcal{L}_{\text{CE-Uni}} = \sum_{m=1}^{M} \mathcal{L}_{\text{CE}}(\mathbf{y}, \hat{\mathbf{y}}^{(m)}) \tag{14}$$

We merge the unimodal latent representations by concatenating them and pass the combined representation to the output classifier. These predictions serve as the final outputs $\hat{\mathbf{y}}_f$ used during inference. The multimodal prediction process aims to minimize the cross-entropy loss between $\hat{\mathbf{y}}_f$ and the corresponding labels. The multimodal cross-entropy loss is:

$$\mathcal{L}_{\text{CE-Multi}} = \mathcal{L}_{\text{CE}}(\mathbf{y}, \hat{\mathbf{y}}_f) \tag{15}$$

**Combined Learning Objective:** Our overall loss objective utilizes a schedule to combine our M3Co and MultiSClip loss functions weighted by a hyperparamater $\beta$, along with the unimodal and multimodal cross-entropy losses. We use M3Co for the first one-third (Liu et al., 2023) part of training, and then transition to MultiSClip as over-training with a Mixup-based loss can potentially harm generalization. The end-to-end loss is defined as:

$$\mathcal{L}_{\text{Total}} = \beta \cdot \mathcal{L}_{\text{M3Co | MultiSClip}} + \mathcal{L}_{\text{CE-Uni}} + \mathcal{L}_{\text{CE-Multi}} \tag{16}$$

## 3  Experiments

**Datasets.** We evaluate our approach on four diverse publicly available multimodal classification datasets: N24News (Wang et al., 2022), Food-101 (Wang et al., 2015), ROSMAP (Wang et al., 2021a), and BRCA (Wang et al., 2021a). N24News and Food-101 are both bimodal image-text classification datasets. **Food-101** is a food classification dataset, where each sample is linked with a recipe description gathered from web pages and an associated image. **N24News** is a news classification dataset consisting of four text types (Abstract, Caption, Heading, and Body) along with the corresponding images. Following other works (Zou et al., 2023), we use the first three text types for our experiments. ROSMAP and BRCA are publicly available multimodal medical datasets, each containing three modalities: DNA methylation, miRNA expression, and mRNA expression. **ROSMAP** is an Alzeihmer's diagnosis dataset, while **BRCA** is used for breast invasive carcinoma PAM50 subtype classification. Appendix A.3 provides information about the train-val-test splits.

**Evaluation Metrics.** The evaluation metric used for N24News and Food-101 is classification accuracy (ACC). For BRCA, we report accuracy (ACC), macro-averaged F1 score (MF1), and weighted F1 score (WF1). For ROSMAP, we use accuracy (ACC), area under the ROC curve (AUC), and F1 score (F1) as the evaluation metrics.

**Implementation Details.** We use a ViT (pre-trained on the ImageNet-21k dataset) (Dosovitskiy et al., 2020) as the image encoder for N24News and Food-101. For N24News, the text encoder is a pretrained BERT/RoBERTa (Devlin et al., 2018; Zhuang et al., 2021), while we use a pretrained BERT as the text encoder for Food-101. The classifiers for the above two datasets are three layer MLPs with ReLU activations. For ROSMAP and BRCA, which are small datasets, we use two layer MLPs as feature encoders for each modality, and two layer MLPs with ReLU activations as classifiers. The hyperparameter settings and all other details are given in Appendix A.1.

**Hyperparameters Tuning.** To determine the optimal hyperparameters, we conducted experiments by varying $\alpha$ (the Mixup coefficient, where $\lambda$ is sampled from a Beta distribution, Beta$(\alpha, \alpha)$), $\beta$ (the contrastive loss weighting factor), and the M3Co Ratio (the point at which the loss transitions from M3Co to MultiSClip) on the ROSMAP dataset. As inferred from Tables 8, 9, and 10 in Appendix A.2, the optimal values were found to be $\alpha = 0.15$, $\beta = 0.1$, and M3Co Ratio $= 0.33$, aligning with prior findings (Liu et al., 2023; Zhang et al., 2017) on Mixup augmentation in unimodal settings. Deviations from these values resulted in performance degradation.

**Baselines.** We compare our method with various multimodal classification approaches (Van De Wiel et al., 2016; Wang et al., 2021a; Han et al., 2020; Abavisani et al., 2020; Han et al., 2022; Zou et al., 2023; Liang et al., 2022b; Kiela et al., 2019; 2018; Wang et al., 2022; Vielzeuf et al., 2018; Arevalo et al., 2017; Li et al., 2019; Huang et al., 2020; Kim et al., 2021; Narayana et al., 2019; Liu et al., 2021; Hong et al., 2020; Huang et al., 2021; Singh et al., 2019; Wang et al., 2024). Some methods (Kiela et al., 2019; Vielzeuf et al., 2018; Arevalo et al., 2017) focus on integrating global features from individual modality-specific backbones to enhance classification. Others (Li et al., 2019; Kim et al., 2021; Huang et al., 2020; Narayana et al., 2019) use sophisticated pre-trained architectures fine-tuned for specific tasks. UniS-MMC (Zou et al., 2023), the previous state-of-the-art on Food-101 and N24News, uses contrastive learning to align features across modalities with supervision from unimodal predictions. Similarly, Dynamics (Han et al., 2022), the previous state-of-the-art on ROSMAP and BRCA, applies a dynamic multimodal classification strategy. On Food-101 and N24News, we compare against baseline unimodal networks (ViT and BERT/RoBERTa) and our UniConcat baseline, where pre-trained image and text encoders are fine-tuned independently, and the unimodal representations

are simply concatenated for classification. These are typical baselines used in multimodal classification tasks. Detailed baseline descriptions are discussed in Appendix A.8.

## 4 Results

### 4.1 Comparison with Baselines

The results are reported as the average and standard deviation over three runs on Food-101/N24News, and five runs on ROSMAP/BRCA. The best score is highlighted in bold, while the second-best score is underlined. The classification accuracy on N24News and Food-101 are displayed in Table 1 and 3 respectively. In the result tables, **ALI** denotes alignment (indicating if the method employs a contrastive component), while **AGG** specifies whether aggregation is early (combining unimodal feature) or late fusion (combining unimodal decisions).

The experimental results from Table 1, 2, 3, reveal the following findings: **(i)** M3CoL consistently outperforms all SOTA methods across all text sources on N24News when using the same encoders, beats SOTA on all evaluation metrics on ROSMAP and BRCA, and also achieves competitive results on Food-101; **(ii)** contrastive-based methods with any form of alignment demonstrate superior performance compared to other multimodal methods; **(iii)** our proposed M3CoL method, which employs a contrastive-based approach with shared alignment, improves over the traditional contrastive-based models and the latest SOTA multimodal methods; **(iv)** our method augments training samples by creating convex mixtures of raw data points, leading to better and more stable performance through improved contrastive representations, as supported by i-mix (Lee et al., 2020). The marginal improvement in N24News (49K samples) and the slight underperformance in Food-101 (60K samples) suggest that the benefits of M3CoL are more pronounced in low-sample regimes, where data augmentation and contrastive learning play a critical role. This aligns with the nature of the datasets, as smaller datasets often benefit more from structured noise and shared relation learning. We visualize the unimodal and combined representation distribution of our proposed method using UMAP plots in Figure 7 in Appendix A.6.

| Method | Fusion | | Backbone | | ACC ↑ | | |
|---|---|---|---|---|---|---|---|
| | AGG | ALI | Image | Text | Headline | Caption | Abstract |
| Image-only | - | - | ViT | - | 54.1 | *(no text source used)* | |
| Text-only | - | - | - | BERT | 72.1 | 72.7 | 78.3 |
| UniConcat | Early | ✗ | ViT | BERT | 78.6 | 76.8 | 80.8 |
| UniS-MMC | Early | ✓ | ViT | BERT | 80.3 | 77.5 | 83.2 |
| M3CoL (Ours) | Early | ✓ | ViT | BERT | $80.8_{\pm0.05}$ | $78.0_{\pm0.03}$ | $83.8_{\pm0.06}$ |
| Text-only | - | - | - | RoBERTa | 71.8 | 72.9 | 79.7 |
| UniConcat | Early | ✗ | ViT | RoBERTa | 78.9 | 77.9 | 83.5 |
| N24News | Early | ✗ | ViT | RoBERTa | 79.41 | 77.45 | 83.33 |
| UniS-MMC | Early | ✓ | ViT | RoBERTa | 80.3 | 78.1 | 84.2 |
| M3CoL (Ours) | Early | ✓ | ViT | RoBERTa | $80.9_{\pm0.19}$ | $79.2_{\pm0.08}$ | $84.7_{\pm0.03}$ |

Table 1: Accuracy (ACC) on N24News on three text sources. AGG denotes early/late modality fusion, ALI indicates presence/absence of alignment. Our method consistently outperforms SOTA across all text sources and backbone combinations. Baseline details are provided in Appendix A.8.

### 4.2 Analysis of Our Method

**Effect of Vanilla Mixup.** Mixup involves two main components: the random convex combination of raw inputs and the corresponding convex combination of one-hot label encodings. To assess the performance of our M3CoL method in comparison to this Mixup strategy, we conduct experiments on Food-101 and N24News (text source: abstract). We remove the contrastive loss from our framework (Eq. 16) while keeping

| Method | Fusion | | ROSMAP | | | BRCA | | |
|---|---|---|---|---|---|---|---|---|
| | AGG | ALI | ACC ↑ | F1 ↑ | AUC ↑ | ACC ↑ | WF1 ↑ | MF1 ↑ |
| GRidge | Early | ✗ | 76.0 | 76.9 | 84.1 | 74.5 | 72.6 | 65.6 |
| BPLSDA | Early | ✗ | 74.2 | 75.5 | 83.0 | 64.2 | 53.4 | 36.9 |
| BSPLSDA | Early | ✗ | 75.3 | 76.4 | 83.8 | 63.9 | 52.2 | 35.1 |
| MOGONET | Late | ✗ | 81.5 | 82.1 | 87.4 | 82.9 | 82.5 | 77.4 |
| TMC | Late | ✗ | 82.5 | 82.3 | 88.5 | 84.2 | 84.4 | 80.6 |
| CF | Early | ✗ | 78.4 | 78.8 | 88.0 | 81.5 | 81.5 | 77.1 |
| GMU | Early | ✗ | 77.6 | 78.4 | 86.9 | 80.0 | 79.8 | 74.6 |
| MOSEGCN | Early | ✗ | 83.0 | 82.7 | 83.2 | 86.7 | 86.8 | 81.1 |
| DYNAMICS | Early | ✗ | 85.7 | 86.3 | 91.1 | 87.7 | 88.0 | 84.5 |
| M3CoL (Ours) | Early | ✓ | $88.7_{\pm0.94}$ | $88.5_{\pm0.94}$ | $92.6_{\pm0.59}$ | $88.4_{\pm0.57}$ | $89.0_{\pm0.42}$ | $86.2_{\pm0.54}$ |

Table 2: Comparison of Accuracy (ACC), Area Under the Curve (AUC), F1 score (F1) on ROSMAP, and Accuracy (ACC), Weighted F1 score (WF1), and Micro F1 score (MF1) on BRCA datasets. AGG denotes early/late modality fusion, ALI indicates presence/absence of alignment. Our method significantly outperforms SOTA across all metrics. Baseline details are provided in Appendix A.8.

| Method | Fusion | | Backbone | | ACC ↑ |
|---|---|---|---|---|---|
| | AGG | ALI | Image | Text | |
| Image-only | - | - | ViT | - | 73.1 |
| Text-only | - | - | - | BERT | 86.8 |
| UniConcat | Early | ✗ | ViT | BERT | 93.7 |
| MCCE | Early | ✗ | DenseNet | BERT | 91.3 |
| CentralNet | Early | ✗ | LeNet5 | LeNet5 | 91.5 |
| GMU | Early | ✗ | RNN | VGG | 90.6 |
| ELS-MMC | Early | ✗ | ResNet-152 | BOW features | 90.8 |
| MMBT | Early | ✗ | ResNet-152 | BERT | 91.7 |
| HUSE | Early | ✓ | Graph-RISE | BERT | 92.3 |
| VisualBERT | ✗ | ✓ | FasterRCNN+BERT | BERT | 92.3 |
| PixelBERT | Early | ✓ | ResNet | BERT | 92.6 |
| ViLT | Early | ✓ | ViT | BERT | 92.9 |
| CMA-CLIP | Early | ✓ | ViT | BERT | 93.1 |
| ME | Early | ✗ | DenseNet | BERT | 94.7 |
| UniS-MMC | Early | ✓ | ViT | BERT | 94.7 |
| M3CoL (Ours) | Early | ✓ | ViT | BERT | $94.3_{\pm0.04}$ |

Table 3: Accuracy (ACC) comparison on Food-101. AGG denotes early/late modality fusion, ALI indicates presence/absence of alignment. Baseline details are provided in Appendix A.8.

the rest of the modules unchanged. Table 4 shows that the **Mixup** technique underperforms relative to our proposed M3CoL approach (Testing accuracy curves shown in Figure 6a). The observed accuracy gap can be attributed to excessive noise introduced by label mixing, and the lack of a contrastive approach with an alignment component. This indicates that the vanilla Mixup strategy introduces additional noise which impairs the model's ability to learn effective representations, while our M3CoL framework benefits from the structured contrastive approach.

**Effect of Loss & Unimodality Supervision.** To assess the necessity of each component in the framework, we investigate several design choices: (i) the framework's performance without the supervision of unimodal modules during training, and (ii) the performance differences between using only MultiSClip and only M3Co

| Method | ACC ↑ | | | |
|---|---|---|---|---|
| | ROSMAP | BRCA | Food-101 | N24News |
| Mixup | $84.13_{\pm 0.74}$ | $84.52_{\pm 0.46}$ | $93.14_{\pm 0.02}$ | $81.57_{\pm 0.24}$ |
| M3CoL (No Unimodal Supervision) | $85.14_{\pm 0.85}$ | $86.93_{\pm 0.52}$ | $94.12_{\pm 0.02}$ | $84.26_{\pm 0.11}$ |
| M3CoL (only MultiSClip) | $86.84_{\pm 0.34}$ | $87.38_{\pm 0.41}$ | $94.23_{\pm 0.01}$ | $84.06_{\pm 0.18}$ |
| M3CoL (only M3Co) | $87.42_{\pm 0.63}$ | $87.74_{\pm 0.42}$ | $94.24_{\pm 0.12}$ | $84.57_{\pm 0.08}$ |
| M3CoL (0.33 M3Co + 0.67 MultiSClip) | $88.67_{\pm 0.94}$ | $88.38_{\pm 0.57}$ | $94.27_{\pm 0.04}$ | $84.72_{\pm 0.03}$ |

Table 4: Accuracy (ACC) on ROSMAP, BRCA, N24News, and Food-101 datasets under different settings of our method. For N24News, source: abstract and encoder: RoBERTa.

loss during end-to-end training. The M3CoL (**No Unimodal Supervision**) result indicates that excluding the unimodal prediction module results in a decline in performance as shown in Table 4 and Figure 6a, highlighting its importance as it allows the model to compensate for the weaknesses of one modality with the strengths of another. Additionally, the M3Co loss (**only M3Co**) outperforms the MultiSClip loss (**only MultiSClip**) by learning more robust representations through Mixup-based techniques, which prevent trivial discrimination of positive pairs. Furthermore, using an individual contrastive alignment approach (**only M3Co**) throughout the entire training process without transitioning to the MultiSClip loss results in suboptimal outcomes. This can be attributed to the risk of over-training with Mixup-based loss, which may negatively impact generalization. This demonstrates the necessity of the transition of the contrastive loss during training (**0.33 M3Co + 0.67 MultiSClip**). Figure 6b displays the accuracy plots on the N24News dataset, for these losses. In addition to the ACC scores presented in Table 4, we also report the performance of other metrics, where available, on the ROSMAP and BRCA datasets under various settings of our method, as shown in Table 5.

| Method | ROSMAP | | BRCA | |
|---|---|---|---|---|
| | F1 ↑ | AUC ↑ | WF1 ↑ | MF1 ↑ |
| Mixup | $84.45_{\pm 0.48}$ | $88.73_{\pm 0.53}$ | $84.66_{\pm 0.48}$ | $82.88_{\pm 0.38}$ |
| M3CoL (No Unimodal Supervision) | $86.27_{\pm 0.39}$ | $90.20_{\pm 0.82}$ | $86.92_{\pm 0.36}$ | $85.08_{\pm 0.37}$ |
| M3CoL (only MultiSClip) | $86.76_{\pm 0.58}$ | $90.75_{\pm 0.49}$ | $87.41_{\pm 0.47}$ | $85.48_{\pm 0.35}$ |
| M3CoL (only M3Co) | $87.54_{\pm 0.62}$ | $91.41_{\pm 0.54}$ | $88.06_{\pm 0.41}$ | $85.82_{\pm 0.34}$ |
| M3CoL (0.33 M3Co + 0.67 MultiSClip) | $88.51_{\pm 0.94}$ | $92.62_{\pm 0.59}$ | $89.02_{\pm 0.42}$ | $86.20_{\pm 0.54}$ |

Table 5: Comparison of F1 score (F1), Area Under the Curve (AUC) on ROSMAP, and Weighted F1 score (WF1), Micro F1 score (MF1) on BRCA, under different settings of our method.

**Visualization of Attention Heatmaps.** The attention heatmaps generated using the embeddings from our trained M3CoL model in Figure 3 and 4 highlight image regions most relevant to the input word. We generate text embeddings for class label words and corresponding image patch embeddings, computing attention scores as their dot product. This visualization aids in understanding the model's focus, decision-making process, and association between class labels and specific image regions. Importantly, it also indicates the correctness of the learned multimodal representations, revealing the model's ability to learn shared relations amongst different modalities, and ground visual concepts to semantically meaningful regions.

**Testing on Random Data and Single-Corrupt Modalities.** To showcase the benefits of our framework over traditional contrastive methods, we evaluate the impact of incorporating Mixup-based contrastive loss (M3Co) during training, highlighting its improvements over standard approaches. It is well-established that deep networks tend to exhibit overconfidence, particularly when making predictions on random or adversarial inputs (Hendrycks & Gimpel, 2016). Previous research has demonstrated that Mixup can mitigate this issue, and our goal is to validate its effectiveness in this context (Thulasidasan et al., 2019). We evaluate

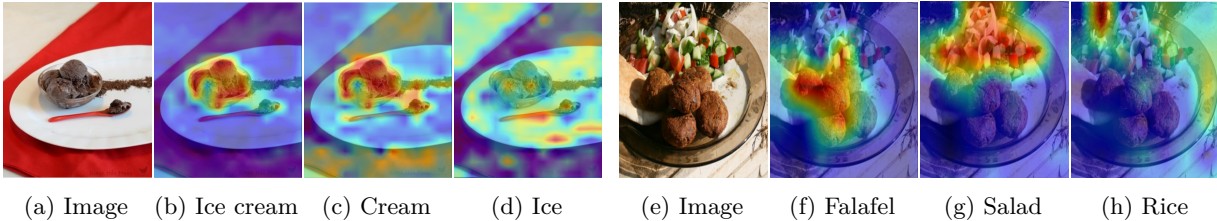

| (a) Image | (b) Ice cream | (c) Cream | (d) Ice | (e) Image | (f) Falafel | (g) Salad | (h) Rice |

Figure 3: Text-guided visual grounding with varying input prompts. (a, e) Original images. (b-d) Attention heatmaps for "ice cream" class. (f-h) Heatmaps for "falafel" class. Ice cream example: (b) "Ice cream": Concentrated focus on ice cream, (c) "Cream": Maintained but diffused focus, (d) "Ice": Dispersed attention. Falafel example: (f) "Falafel": Localized focus on falafel, (g) "Salad": Attention shift to salad component, (h) "Rice": Minimal attention (absent in image). Warmer colors indicate higher attention scores.

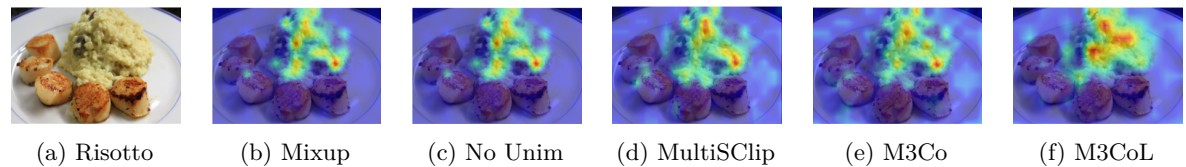

| (a) Risotto | (b) Mixup | (c) No Unim | (d) MultiSClip | (e) M3Co | (f) M3CoL |

Figure 4: Text-guided visual grounding with ablated model variations. (a) Original image. (b-f) Attention heatmaps generated using text embedding (class name: "Risotto") and patch embeddings for different variations of the model. Our proposed M3CoL model (f) demonstrates superior attention localization compared to ablated versions (b-e), corroborating the quantitative results presented in Table 4. Warmer colors indicate higher attention scores. (Here, No Unim: No Unimodal Supervision)

the confidence scores produced using M3CoL (0.33 M3Co + 0.67 MultiSClip) loss in comparison to only MultiSClip loss when predicting on random noise images and text encoder outputs. Our results show that the model trained with M3CoL exhibits lower confidence in its predictions when both modalities are replaced with random inputs. This demonstrates that incorporating M3CoL enhances the reliability of predictions, especially in the presence of corrupted or random inputs.

To evaluate the robustness of our approach, we conduct experiments where one input modality was corrupted with random noise. Table 6 compares the performance of M3CoL (0.33 M3Co + 0.67 MultiSClip) against only MultiSClip under these conditions. Our M3CoL method demonstrates superior robustness to modality corruption, consistently outperforming MultiSClip. For image corruption, we substituted the original images

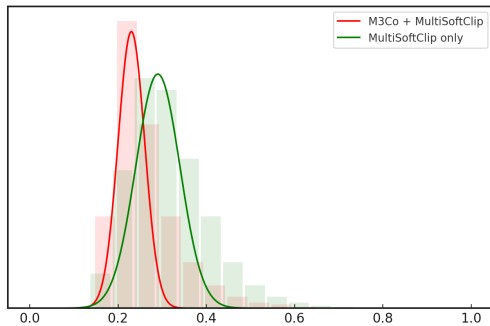

Figure 5: N24News - Confidence scores when tested on random inputs.

with random noise sampled from a Gaussian distribution, parameterized to match the mean and variance of the training set. Similarly, for text corruption, we replaced the original text embeddings with random outputs from the text encoder, again following a Gaussian distribution with statistics matching the training

| Method | Modality Corrupted | | |
|--------|------|-------|------|
| | None | Image | Text |
| MutliSClip | 84.06 | 76.06 | 46.91 |
| M3CoL | 84.72 | 77.24 | 47.94 |

Table 6: N24News - Accuracy when tested on data with modality corruption.

data. For both the above experiments, we use the N24News dataset, with the abstract as the text source and a RoBERTa-based text encoder.

**Error Analysis.** To evaluate the efficacy of our multimodal approach in integrating and leveraging image and text features, we performed a comprehensive error analysis, comparing it with image-only (ViT) and text-only (RoBERTa) models using the N24News dataset (see Table 12 in Appendix A.5). The analysis reveals that our method excels when both modalities are correctly classified (42.71:0.03 correct-to-incorrect ratio). This shows that our model can learn valuable insights from the fusion of image and text features, which may not be discovered when processing them separately. In cases where only one modality is correctly classified, our model effectively leverages the accurate modality (27.77+8.11=35.88):(1.29+3.25=4.54) correct-to-incorrect ratio. As shown in Table 12, our method excels not only when both modalities are correctly classified but also demonstrates strong performance even when both are misclassified, indicating effective feature fusion. In addition, it successfully leverages information when only one modality is classified correctly. This analysis demonstrates our method's robustness and highlights its superiority over unimodal approaches.

## 5 Related Work

**Contrastive Learning.** Contrastive learning has driven significant progress in unimodal and multimodal representation learning by distinguishing between similar (positive) and dissimilar (negative) pairs. In multimodal contexts, cross-modal contrastive techniques align representations from different modalities (Radford et al., 2021; Jia et al., 2021; Kamath et al., 2021), with approaches like CrossCLR (Zolfaghari et al., 2021) and GMC (Poklukar et al., 2022) focusing on global and modality-specific representations. Contrastive learning approaches for paired image-text data, such as CLIP (Radford et al., 2021), ALIGN (Jia et al., 2021), and BASIC (Pham et al., 2023), have demonstrated remarkable success across diverse vision-language tasks. Subsequent works have aimed to enhance the efficacy and data efficiency of CLIP training, incorporating self-supervised techniques (SLIP (Mu et al., 2022), DeCLIP (Li et al., 2021)) and fine-grained alignment (FILIP (Li et al., 2023)). The CLIP framework relies on data augmentations to prevent overfitting and the learning of ineffective shortcuts (Chen et al., 2020; Robinson et al., 2021), a common practice in contrastive learning.

**Unimodal and Multimodal Data Augmentation.** Data augmentation has been integral to the success of deep learning, especially for small training sets. In computer vision, techniques have evolved from basic transformations to advanced methods like Cutout (DeVries & Taylor, 2017), Mixup (Zhang et al., 2017), CutMix (Yun et al., 2019), and automated approaches (Cubuk et al., 2019; 2020). NLP augmentation includes paraphrasing, token replacement (Zhang et al., 2015; Jiao et al., 2019), and noise injection (Yan et al., 2019). Multimodal data augmentation, primarily focused on vision-text tasks, has seen limited exploration, with approaches including back-translation for visual question answering (Tang et al., 2020), text generation from images (Wang et al., 2021b), and external knowledge querying for cross-modal retrieval (Gur et al., 2021). MixGen (Hao et al., 2023) generates new image-text pairs through image interpolation and text concatenation. In contrast, our proposed augmentation technique focusing on the early training phase is fully automatic, applicable to arbitrary modalities, and designed to leverage inherent shared relations in multimodal data.

**Relation to Mixup.** Mixup (Zhang et al., 2017), a pivotal regularization strategy, enhances model robustness and generalization by generating samples through convex combinations of existing data points. Originally introduced for computer vision, it has been adapted to NLP by applying the technique to text embeddings (Guo et al., 2019). Our proposed augmentation differs from Mixup in several key aspects: it is designed for multimodal data, takes inputs from different modalities, and does not rely on one-hot label encodings. By

extending the Mixup paradigm to complex, multimodal scenarios and focusing on the early training phase, our method broadens its applicability while leveraging inherent shared relations in multimodal data.

**Relation to Previous Work.** Our method builds upon and extends several key ideas from recent advancements in multimodal learning. M3CoL shares the use of contrastive learning for multimodal alignment but goes beyond by introducing Mixup-based augmentation to capture shared relations across modalities, which UniS-MMC (Zou et al., 2023) does not address. While Dynamics (Han et al., 2020) focuses on dynamic fusion for trustworthy classification, M3CoL emphasizes learning robust representations through Mixup-based contrastive loss, enhancing generalization and interpretability. SoftCLIP (Gao et al., 2024) introduces softer cross-modal alignment, whereas M3CoL utilizes Mixup to create structured noise and align mixed samples, capturing nuanced shared relations. MixGen (Hao et al., 2023) explores multimodal data augmentation through image interpolation and text concatenation. M3CoL differs by focusing on Mixup-based contrastive learning to align mixed samples, which is more targeted at capturing shared relations rather than generating new data points. M3CoL advances multimodal learning by addressing the limitations of traditional contrastive methods and introducing a novel approach to capturing shared relations.

## 6  Discussion and Limitations

**Discussion and Conclusions**. We propose M3Co, a novel contrastive-based alignment method that captures shared relations beyond explicit pairwise associations by aligning mixed samples from one modality with corresponding samples from others. The M3Co loss, combined with an architecture leveraging unimodal and fusion modules, enables continuous updating of representations necessary for accurate predictions and deeper integration of modalities. This method generalizes across diverse domains, including image-text, high-dimensional multi-omics, and data with more than two modalities. Experiments on four public multimodal classification datasets demonstrate the effectiveness of our approach in learning robust representations that surpass traditional multimodal alignment techniques.

The computation cost in loss calculation in M3CoL scales as $\mathcal{O}(2M^2N^2)$, which is only a factor of 2 larger than conventional contrastive learning $\mathcal{O}(M^2N^2)$. This is because, in a batch of size $N$, each sample is mixed pairwise with one other sample (e.g., image 1 with image $N$, image 2 with image $N-1$, etc.), and the resulting mixture creates two positive relations with corresponding samples from other modalities. While the computational complexity scales quadratically with both the number of modalities ($M^2$) and the batch size ($N^2$), the overall overhead is not significantly higher than conventional contrastive approaches. We did not observe significant differences in training times compared to conventional contrastive learning. Additionally, the Mixup-based augmentation is applied only during the first one-third of training, not during evaluation, which helps manage computational overhead.

**Analysis of Learned Representations.** Aligning representations across modalities presents significant challenges due to the complex, often non-bijective relationships in real-world multimodal data. These relationships can involve many-to-many mappings or even lack clear associations, as exemplified by linguistic ambiguities and synonymy in vision-language tasks (Liang et al., 2022a). Our approach incorporates Mixup-based contrastive learning, introducing controlled noise that mirrors the inherent variability in multimodal data, thus enhancing robustness and generalizability. The visualization of attention heatmaps (Figure 3 and Figure 4) shows that M3CoL learns semantically meaningful multimodal representations by grounding visual concepts to relevant image regions. When tested with corrupted or random inputs, M3CoL demonstrates lower confidence scores (Table 6), reducing overconfidence and enhancing reliability.

**Limitations and Future Work.** M3CoL demonstrates promising results, yet faces optimization challenges due to the inherent limitations of multimodal frameworks, particularly extended training times on large-scale datasets like Food-101 and involves multiple hyperparameters, making it more complex to identify the optimal configuration. The method's modality-agnostic nature and effective use of mixup augmentation suggest its potential adaptability to various multimodal tasks, especially where data augmentation and learning real-world inter-modal relationships are crucial. Future work should focus on investigating domain adaptation strategies, validating M3CoL's utility on downstream tasks such as visual question answering and information retrieval, and enhancing interpretability through explainable AI techniques. These advancements, coupled with comprehensive hyperparameter tuning, will likely broaden M3CoL's impact in multimodal research.

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

# A  Appendix

## A.1  Experimental Details

The models were trained on either an NVIDIA RTX A6000 or an NVIDIA A100-SXM4-80GB GPU. The results are reported as the average and standard deviation over three runs on Food-101 and N24News, and five runs on ROSMAP and BRCA. We use a grid search on the validation set to search for optimal hyperparameters. The temperature parameter for the M3Co and MultiSClip losses is set to 0.1. The corresponding loss coefficient $\beta$ is 0.1 to keep the loss value in the same range as the other losses. We use the Adam optimizer (Kingma & Ba, 2014) for all datasets. For Food-101 and N24News, the learning rate scheduler is ReduceLROnPlateau with validation accuracy as the monitored metric, lr factor of 0.2, and lr patience of 2. For ROSMAP and BRCA, we use the StepLR scheduler with a step size of 250. For Food-101 and N24News, the maximum token length of the text input for the BERT/RoBERTa encoders is 512. Other hyperparameter details are provided in Table 7.

| Hyperparameter | N24News | Food-101 | ROSMAP | BRCA |
|---|---|---|---|---|
| Embedding dimension | 768 | 768 | 1000 | 768 |
| Classifier dimension | 256 | 256 | 1000 | 768 |
| Learning rate | $10^{-4}$ | $10^{-4}$ | $5 \cdot 10^{-3}$ | $5 \cdot 10^{-3}$ |
| Weight decay | $10^{-4}$ | $10^{-4}$ | $10^{-3}$ | $10^{-3}$ |
| Batch size | 32 | 32 | - | - |
| Batch gradient | 128 | 128 | - | - |
| Dropout (classifier) | 0 | 0 | 0.5 | 0.5 |
| Epochs | 50 | 50 | 500 | 500 |

Table 7: Experimental hyperparameter values for our proposed model across all the four datasets.

## A.2  Hyperparameters Selection

We illustrate how performance varies with different hyperparameters and describe our selection process through Tables 8, 9, and 10.

| $\alpha$ | ACC ↑ | F1 ↑ | AUC ↑ |
|---|---|---|---|
| 0.05 | 84.91 | 84.11 | 91.34 |
| 0.15 | **87.74** | **87.62** | **92.98** |
| 0.4 | 85.85 | 85.18 | 91.55 |
| 0.8 | 83.96 | 84.11 | 91.52 |

Table 8: Performance comparison for different $\alpha$ values. $\alpha$ is used to calculate the Mixup coefficient as $\lambda \sim \text{Beta}(\alpha, \alpha)$.

| $\beta$ | ACC ↑ | F1 ↑ | AUC ↑ |
|---|---|---|---|
| 0.05 | 84.91 | 85.72 | 91.77 |
| 0.1 | **87.74** | **87.62** | **92.98** |
| 0.5 | 85.85 | 86.49 | 92.55 |
| 1 | 85.85 | 85.72 | 92.41 |

Table 9: Performance comparison for different $\beta$ values. $\beta$ is the contrastive loss scaling factor.

| M3Co Ratio | ACC ↑ | F1 ↑ | AUC ↑ |
|:---:|:---:|:---:|:---:|
| 0 | 85.85 | 85.98 | 91.66 |
| 0.2 | 85.85 | 85.71 | 92.41 |
| 0.33 | **87.74** | **87.62** | **92.98** |
| 0.5 | 86.79 | 86.79 | 92.58 |
| 0.8 | 86.79 | 86.27 | 92.58 |
| 1 | 86.79 | 86.27 | 92.58 |

Table 10: Performance comparison for different M3Co Ratio values, which decides when to switch from the M3Co loss to the MultiSClip loss.

### A.3 Dataset Information and Splits

The datasets used in our experiments can be downloaded from the following sources: Food-101 from https://visiir.isir.upmc.fr, N24News from https://github.com/billywzh717/N24News, and BRCA and ROSMAP from https://github.com/txWang/MOGONET.

To ensure a fair comparison with previous works, we adopt the default split method detailed in Table 11. As the Food-101 dataset does not include a validation set, we partition 5,000 samples from the training set to create one, which is conistent with other baselines.

| Dataset | Modalities | Modality Types | Train | Validation | Test | Classes |
|:---|:---:|:---:|:---:|:---:|:---:|:---:|
| Food-101 | 2 | Image, text | 60101 | 5000 | 21695 | 101 |
| N24News | 2 | Image, text | 48988 | 6123 | 6124 | 24 |
| ROSMAP | 3 | mRNA, miRNA, DNA | 245 | - | 106 | 2 |
| BRCA | 3 | mRNA, miRNA, DNA | 612 | - | 263 | 5 |

Table 11: Statistics for the four datasets: Food-101, N24News, ROSMAP, and BRCA. Note: miRNA stands for microRNA, and mRNA stands for messenger RNA.

### A.4 Ablation Studies on the N24News Dataset

The accuracy plots for the N24News dataset (text source: abstract, text encoder: RoBERTa) are used to compare our method and its variants. Our proposed M3CoL approach outperforms the Mixup technique, as shown in Figure 6a. Ablating the unimodal supervision in M3CoL leads to a performance decline, indicating the importance of the unimodal prediction module, as shown in Figure 6a. Furthermore, the M3Co loss achieves better results than the MultiSClip loss. Training solely with either the M3Co loss or the MultiSClip loss alignment approach yields suboptimal performance when compared to their strategic combination, as shown in Figure 6b. The quantitative results are given in Table 4.

### A.5 Error Analysis Table

We provide an in-depth error analysis in Table 12 on the N24News dataset, as discussed in Section 4.2.

### A.6 UMAP Plots

We generate UMAP plots on embeddings derived from the N24News and Food-101 datasets to visualize the clustering performance of our M3CoL model. For each dataset, we randomly select 10 classes and generate the corresponding embeddings from the image encoder, text encoder, and their concatenated multimodal representatio, using our trained M3CoL model. Figure 7 shows that the image embeddings depict less distinct clusters, indicating less effective inter-cluster separation compared to the text encoder embeddings. The

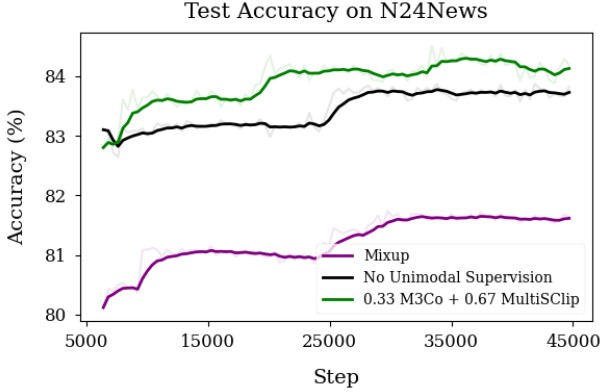 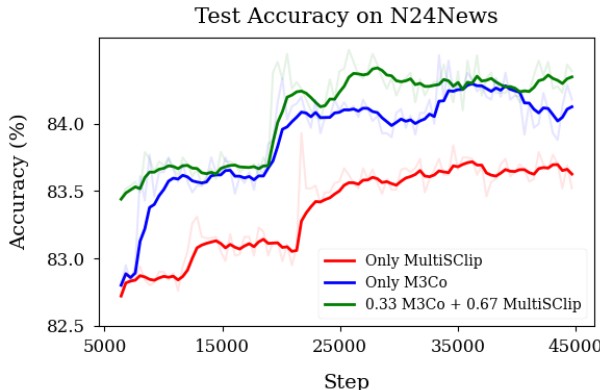

(a) Comparison of M3CoL and its variants using Mixup and No Unimodal Supervision.

(b) Comparison of M3CoL and its variants using only M3Co and only MultiSoftClip loss.

Figure 6: Test accuracy plots showing comparison of M3CoL and its variants on the N24News dataset (text source: abstract, text encoder: RoBERTa).

| Unimodal Prediction | | Multimodal Prediction % | |
|---|---|---|---|
| **Text** | **Image** | **Correct** | **Incorrect** |
| True | True | 42.71 | 0.03 |
| True | False | 27.77 | 1.29 |
| False | True | 8.11 | 3.25 |
| False | False | 2.31 | 14.53 |

Table 12: Error analysis on N24News with text encoder RoBERTa and text source "headline". True and False denote the correctness of the unimodal predictions. Multimodal Prediction % shows the resulting test set ratio of the final predictions.

concatenated embeddings, however, result in the best-defined clusters, suggesting that the final multimodal representations preserve and potentially enhance class-distinguishing features. These observations align with our quantitative results presented in Table 1 and 3.

### A.7 Additional Visualization Attention Heatmaps

Following Section 4.2, we generate heatmaps using class embeddings and patch embeddings for some more examples in Food-101. These are depicted in Figure 8.

### A.8 Baseline Details

The baselines used in our comparsions are described in details as follows:

- **GRidge** (Van De Wiel et al., 2016) dynamically incorporates multimodal data to adjust regularization penalties, improving predictive accuracy in genomic classification scenarios.

- **BPLSDA** (Block partial least squares discriminant analysis) (Singh et al., 2019) analyzes multimodal data in latent space and **BSPLSDA** (Block sparse partial least squares discriminant analysis) (Singh et al., 2019) adds sparsity constraints to BPLSDA to extract relevant features.

- **MOGONET** (Wang et al., 2021a) integrates GCNs with a View Correlation Discovery Network (VCDN) to process multi-omics data. The initial predictions from each omics-specific GCN are

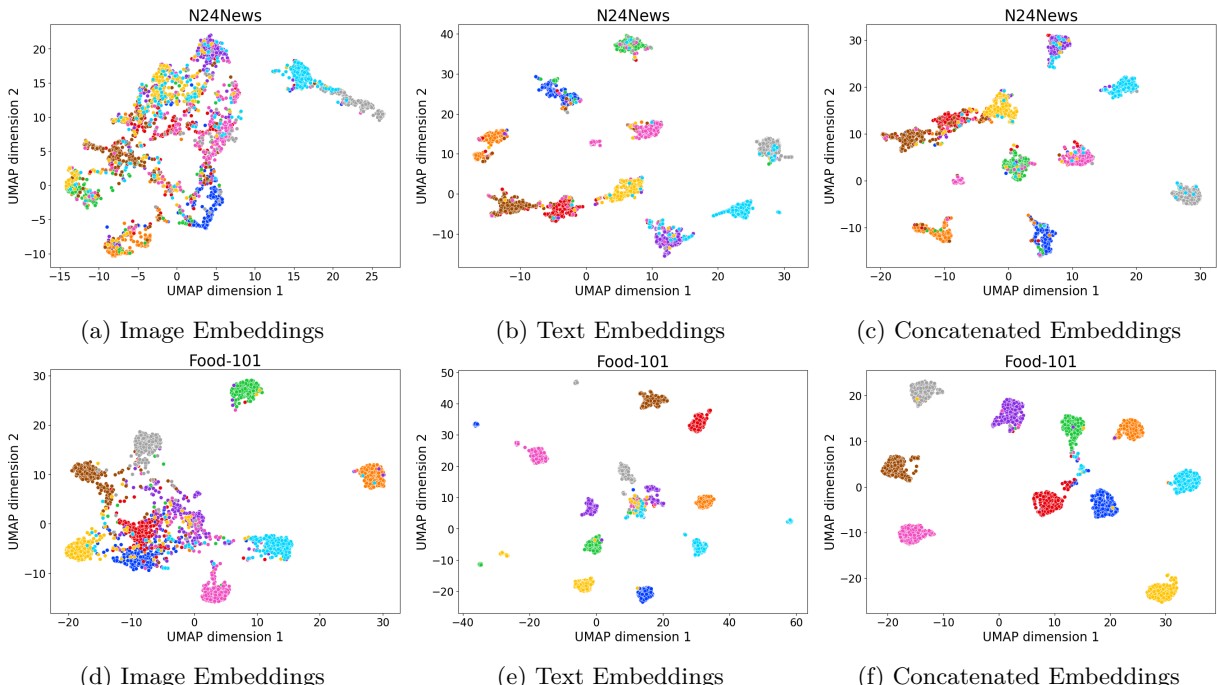

(a) Image Embeddings
(b) Text Embeddings
(c) Concatenated Embeddings

(d) Image Embeddings
(e) Text Embeddings
(f) Concatenated Embeddings

Figure 7: UMAP plots of embeddings from the N24News (source: abstract and encoder: RoBERTa) and Food-101 datasets. We generate UMAP plots for the representations generated by the image encoder, text encoder, and their concatenated multimodal representations, using our trained M3CoL model. Concatenated embeddings exhibit superior clustering, while text embeddings outperform image embeddings. Consistent patterns across datasets demonstrate M3CoL's effectiveness in fusing multimodal information and enhancing semantic representations.

consolidated using the VCDN, which identifies and leverages cross-omics label correlations to improve prediction accuracy.

- **CF** (Concatenation of Final Multimodal Representations) (Hong et al., 2020; Huang et al., 2021) creates representations by combining late stage representations of multiple modalities.

- **TMC** (Han et al., 2020) enhances decision-making by dynamically integrating multiple views based on confidence levels, for robust and reliable fusion.

- **MCCE** (Abavisani et al., 2020) uses DenseNet and BERT for feature extraction then applying stochastic transitions between multi-modal embeddings during training to enhance generalization and to handle sparse data effectively.

- **Dynamics** (Han et al., 2022) presents an approach for trustworthy multi-modal classification, specifically designed for high-stakes environments like medical diagnosis. The model dynamically assesses both feature-level and modality-level informativeness, using a sparse gating mechanism to filter and integrate the most relevant features and modalities per sample.

- **UniS-MMC** (Zou et al., 2023) uses a contrastive learning approach that relies on making unimodal predictions, evaluating the agreement or discrepancy between these predictions and the ground truth, and using this insight to align feature vectors across various modalities through a contrastive loss.

- **ME** (Liang et al., 2022b) leverages cross-modal information by transforming features between modalities. It achieves this by integrating a Multimodal Information Injection Plug-in (MI2P) with pre-trained models, enabling them to process image-text pairs without structural modifications.

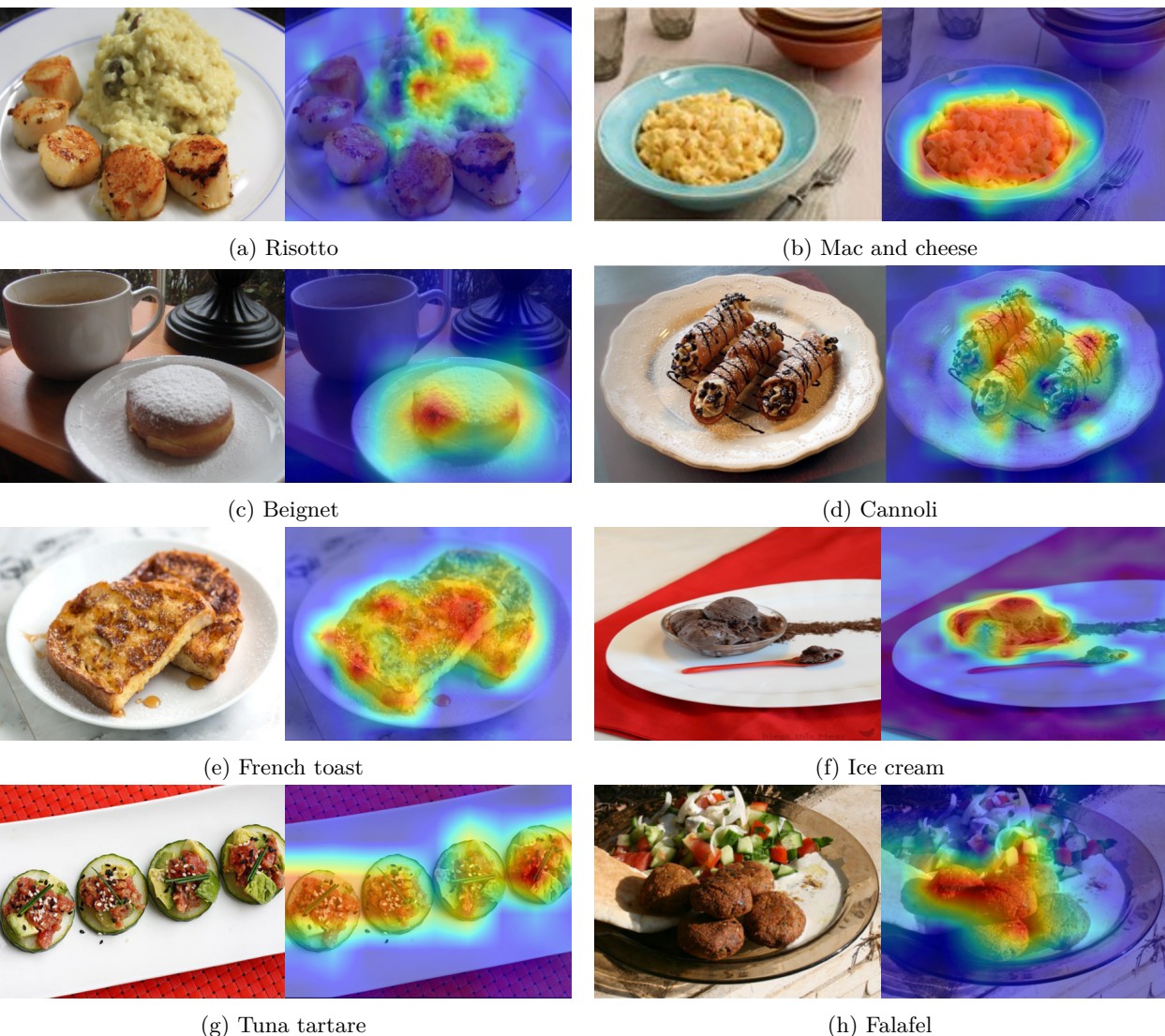

(a) Risotto        (b) Mac and cheese

(c) Beignet        (d) Cannoli

(e) French toast        (f) Ice cream

(g) Tuna tartare        (h) Falafel

Figure 8: Attention heatmaps demonstrating text-guided visual grounding for samples in the Food-101 dataset. Warmer colors indicate higher attention scores.

- **MMBT** (Kiela et al., 2019) leverages the strengths of pre-trained text and image encoders, effectively combining them within a BERT-like framework by mapping image embeddings into the textual token space.

- **ELS-MMC** (Kiela et al., 2018) investigates various multimodal fusion techniques for integrating discrete (text) and continuous (visual) modalities to enhance classification tasks in a resource-efficient manner.

- **N24News** (Wang et al., 2022) presents a novel dataset from The New York Times with text and image data across 24 categories. It utilizes a multitask multimodal strategy, employing ViT for image processing and RoBERTa for text analysis, with features concatenated for final classification.

- **CentralNet** (Vielzeuf et al., 2018) uses separate convolutional networks for each modality, linked via a central network that generates a unified feature representation and also applies multi-task learning to refine and regulate these features.

- **GMU** (Arevalo et al., 2017) employs multiplicative gates that dynamically adjust the influence of each modality on its activation thereby deriving a sophisticated intermediate representation tailored for specific applications.

- **VisualBERT** (Li et al., 2019) employs a series of transformer layers to align textual elements and corresponding image regions through self-attention mechanisms.

- **PixelBert** (Huang et al., 2020) directly aligns image pixels with textual descriptions using a deep multi-modal transformer and establishes a direct semantic connection at the pixel and text level.

- **ViLT** (Kim et al., 2021) implements a BERT-like transformer model that processes visual data in a convolution-free manner, similar to textual data, thereby simplifying input feature extraction and reducing computational demands.

- **HUSE** (Narayana et al., 2019) constructs a shared latent space that aligns image and text embeddings based on their semantic similarity, enhancing cross-modal representation.

- **CMA-CLIP** (Liu et al., 2021) enhances CLIP (Radford et al., 2021) by integrating two cross-modality attention mechanisms: sequence-wise and modality-wise attention. These attention modules refine the relationships between image patches and text tokens, allowing the model to focus on relevant modalities for specific tasks.

- **MOSEGCN** (Wang et al., 2024) utilizes transformer multi-head self-attention and Similarity Network Fusion (SNF) to learn correlations within and among different omics. This information is then fed into a self-ensembling Graph Convolutional Network (SEGCN) for semi-supervised training and testing.

## A.9 Use of Generative AI Models

In this work, we use the following generative AI model:

- Gemini 1.0 Pro (Team et al., 2023) to generate food item images and captions as displayed in Figure 1, which serve as sample representations from the Food-101 dataset.

