# OpenReview forum: "M3CoL: Harnessing Shared Relations via Multimodal Mixup Contrastive Learning for Multimodal Classification"
_TMLR — Accepted by TMLR_

### Review · Reviewer_Wg4t · 2025-02-09

**Summary Of Contributions:**

In this work, the authors introduce a multi-modal architecture to learn the relationships between modalities and learn relationships between samples beyond pairwise-ones. To this end, the authors soften the iid assumption by pairing samples with data-augmented modalities created from their counterpart modalities and randomly picked samples. The contrastive losses are simmetrized for all modalities and samples, combined with unimodal and multimodal classifiers to further guide the training. Empirically, the authors show that their approach is competitive with existing methods, while qualitatively showing to have learnt better relationships across samples, and being more robust to modality noise, than other approaches.

**Audience:**

Yes

**Broader Impact Concerns:**

I do not think there are any concern worth mentioning in the broader impact beyond the additional computational resources required by the proposed approach.

**Claims And Evidence:**

Yes

**Requested Changes:**

The paper is lacking clarity in the writing as well as on the experimental details, as described above. These problems would need to be addressed in my view. There are other clarity problems and nitpicks I highlight below.

Nitpicks and typos:
- $M_j^{(1)}$ in figure 1 left should be $M_j^{(2)}$. I also do not understand why using Gemini to generate the samples, instead of taking two from the Food101 dataset.
- The dot product with "$L^2$-normalized" vector is the cosine similarity, the authors should say the use that instead. Similarly with all the softmax operations written explicitly everywhere.
- Indexing in the context can by confusing. I would encourage authors to pick a consistent way of using the $i$, $j$, and $k$ subindices and stick to that during the entire manuscript.

**Strengths And Weaknesses:**

**Strengths**
1. The idea of relaxing the iid assumption through data-augmentation is quite interesting.
2. The method seems to empirically work well, and the qualitative examples are convincing.
3. Illustrations help a lot to understand the proposed approach.
4. Experiments are quite extensive and it compares with many different baselines. Qualitative results are quite interesting and I think add value to the manuscript.

**Weakness**
1.  More details should be provided in the experimental part, to fully understand the results. For example:
     1. In Table 6, what is the accuracy when none of the modalities are corrupted?
     2. It would be better to use a histogram for Figure 5 (or two). KDE can be quite misleading at times.
     3. It is not clear to me whether the results for the baselines were reproduced by the authors, or taken from somewhere else (maybe I missed it). But I would try to give enough details to convince the reader that comparisons are fair, which is not the case for me right now.
     4. I would try to explain more in detail the Error Analysis and what it means. As of now, it feels that it has been overly summarized.
     5. I find tables like table 3 too misleading, as they do a poor job at summarizing each method. Trying to obtain any kind of conclusions on "which elements work better" is a bad idea, as many nuances of "seemingly equal models" are hidden to the reader.
     6. The use of bold is also misleading to my eye. There is no statistical significance on the difference between the top methods in many cases.
2. The proposed method is rather complicated as it includes many moving parts, which I am not sure how much tuning they would need in other settings. Moreover, it scales as $O(M^2N^2)$, so it cannot include many modalities or samples. There is no mention as far as I can see on training times.
3. Claims seem to strong in general. For example, it is said that the method is "modality-agnostic", but then it is said that the text modality needs to be treated differently for the mixing.
4. Writing could be improved when explaining the maths, for example, I would completely drop the notation $\tilde p_i^{(1)}$ and use $\tilde p_{i,j}^{(1)}$ everywhere, as it is confusing otherwise.

---

> ### Author Response · Authors · 2025-03-12
> **Comment**
>
> We sincerely thank the reviewer for recognizing the strengths of this work. We greatly appreciate the positive feedback on our interesting idea, its convincing nature, the extensive quantitative experiments, and the strong visual illustrations. We have addressed each identified weakness in a detailed, point-by-point format. The updated manuscript highlights these changes in red.
>
>
>     In Table 6, what is the accuracy when none of the modalities are corrupted?
>
> We have added the accuracy numbers from Tables 1 and 4 for completeness. For N24 News (text source: abstract, encoder: RoBERTa), the accuracy when none of the modalities are corrupted is as follows: MultiSClip - 84.06 and M3CoL - 84.72. We have updated Table 6 accordingly in the uploaded revised manuscript.
>
>     It would be better to use a histogram for Figure 5 (or two). KDE can be quite misleading at times.
>
> Thanks for the suggestion. Please see the updated Figure 5, which includes histograms of the confidence scores when tested on random inputs, as suggested, in the uploaded revised PDF.
>
>     It is not clear to me whether the results for the baselines were reproduced by the authors, or taken from somewhere else (maybe I missed it). But I would try to give enough details to convince the reader that comparisons are fair, which is not the case for me right now.
>
> The results for the baselines were taken from the respective papers, as detailed in Appendix A.7 (Baseline Details). We ensured fair comparisons by using the same evaluation protocols, datasets, and metrics as reported in the original works.
>
>     I would try to explain more in detail the Error Analysis and what it means. As of now, it feels that it has been overly summarized.
>
> We have expanded the error analysis in the main text to provide greater clarity. Our analysis shows that when both modalities (image and text) are correctly classified, our model achieves a high correct-to-incorrect ratio of 42.71:0.03, demonstrating the effectiveness of multimodal fusion. When only one modality is correct, the model leverages the accurate modality, achieving a combined correct-to-incorrect ratio of 35.88:4.54. As shown in Table 12 (in the updated manuscript), our method excels not only when both modalities are correctly classified but also demonstrates strong performance even when both are misclassified, indicating effective feature fusion. Additionally, it successfully utilizes information when only one modality is classified correctly. This analysis highlights our method’s robustness and its superiority over unimodal approaches.
>
>     I find tables like table 3 too misleading, as they do a poor job at summarizing each method. Trying to obtain any kind of conclusions on "which elements work better" is a bad idea, as many nuances of "seemingly equal models" are hidden to the reader.
>
> Table 3 serves to consolidate extensive baseline results from prior works, ensuring a fair comparison. For more detailed descriptions of each baseline, please refer to Appendix A.7 (Baseline Details). To address the effectiveness of individual components, we conducted ablation studies on datasets, demonstrating that each loss (M3Co and MultiSClip) positively contributes to the model's performance. Additionally, saliency maps (Figures 3, 4, and 8) validate that M3Col focuses on semantically relevant regions, further supporting its effectiveness over other variations.
>
>     The use of bold is also misleading to my eye. There is no statistical significance on the difference between the top methods in many cases.
>
> We have removed the bold formatting as suggested to avoid any potential misinterpretation. Regarding statistical significance, we acknowledge that comparisons with some baselines are limited due to the unavailability of variance or significance metrics in their original works. Our results are reported with standard deviations to provide transparency, and we encourage future works to include such details for more robust comparisons.

---

> ### Author Response · Authors · 2025-03-12
> **Comment Continued**
>
> .
>
>     The proposed method is rather complicated as it includes many moving parts, which I am not sure much tuning they would need in other settings. Moreover, it scales as O(M2N2), so it cannot include many modalities or samples. There is no mention as far as I can see on training times.
>
> We acknowledge the concern about complexity and scalability. M3CoL scales as $O(2M^2N^2)$, which is only a factor of 2 larger than conventional contrastive learning $O(M^2N^2)$. This is because, in a batch of size (N), each sample is mixed pairwise with one other sample (e.g., image 1 with image (N), image 2 with image (N-1), etc.), and the resulting mixture creates two positive relations with corresponding samples from other modalities. While the computational complexity scales quadratically with both the number of modalities ($M^2$) and the batch size ($N^2$), the overall overhead is not significantly higher than conventional contrastive approaches. We did not observe significant differences in training times compared to conventional contrastive learning. Additionally, the Mixup-based augmentation is applied only during the first one-third of training, not during evaluation, which helps manage computational overhead. We have included a discussion on computational complexity in Section 6 for clarity.
>
>     Claims seem to strong in general. For example, it is said that the method is "modality-agnostic", but then it is said that the text modality needs to be treated differently for the mixing.
>
> We acknowledge the concern and clarify that while there are two potential approaches for text mixing (raw words or embeddings), mixing at the word level introduces significant complexity due to the many possible strategies and their varying outcomes. To maintain simplicity and focus on the core contribution of our work, we opted for sentence embedding mixing, which has been shown to preserve semantic coherence effectively (Guo et al., 2019). This approach aligns with our goal of modality-agnostic mixing.
>
> Guo et al., (2019) Augmenting Data with Mixup for Sentence Classification: An Empirical Study
>
>     Writing could be improved when explaining the maths, for example, I would completely drop the notation p~i(1) and use p~i,j(1) everywhere, as it is confusing otherwise.
>
> We agree that the notation $\tilde p^1_i$ can be confusing and have revised the text to consistently use $\tilde p^{(1)}_{i,j}$ throughout for clarity. Thank you for the suggestion.
>
>     Mj(1) in figure 1 left should be Mj(2). I also do not understand why using Gemini to generate the samples, instead of taking two from the Food101 dataset.
>
> The first two rows on the left side of the figure indicate a positive relationship, so the index remains the same. In contrast, the last row shows a negative relationship, which is why the index differs. We used Gemini to generate illustrative samples for clarity and simplicity.
>
>     The dot product with "L2-normalized" vector is the cosine similarity, the authors should say the use that instead. Similarly with all the softmax operations written explicitly everywhere.
>
> We will update the text to use cosine similarity instead of using dot product with L2-normalized vectors. Similarly, we will write in the form of softmax operators as well.
>
>     Indexing in the context can by confusing. I would encourage authors to pick a consistent way of using the i, j, and k subindices and stick to that during the entire manuscript.
>
> We will ensure consistent use of subindices (i, j, k) throughout the manuscript to avoid confusion.
>
>
> Thank you for the review once again. The review has helped refine our paper.

---

> > ### Comment · Reviewer_Wg4t · 2025-03-17
> >
> > Thank you for the detailed response, the changes to the manuscript, and the _promised_ changes which I think will improve the readability of this work to invite a broader audience.
> >
> > I have no further questions from my side, but I do share some of the concerns of Reviewer gosT regarding the additional hyperparameters and the potential difficulty to train the model. Please make sure to convince everyone that this is properly addressed.

---

### Review · Reviewer_gosT · 2025-02-10

**Summary Of Contributions:**

The paper introduces M3Col, which is a multimodal modal contrastive framework.

-----------------------
### Method
M3Col consists of the following:
-  First there are two encoders, one for each modality the input to the encoders are embeddings from a pretrained model.
- The encoders take either embedding of the input itself, and the output of the encoder, in this case, is the input to three different classifier heads that try to predict a downstream task.
     - Each of the first two classifier heads takes the input from a single modality equations 1 and  14.
     - The third classifier head gets the concatenation of both modalities equation 15.
- Each encoder also takes the weighted combinations of different samples given by equations 2 and 3; this is the mixup part.
    - For the mixed inputs, the output of the encoders bidirectional Mixup contrastive loss M3Co for each modality given in equations 6,7 and 8.
  - To reduce overfitting that might happen due to a mixup they implement a schedule that transitions from the mixup loss to a non-mixup multimodal contrastive loss given in equations 9,10 and 11.
 - Overall loss is given by equation 16.

-----------------------
### Experiments

The paper tested their approach on 4 different datasets two large ones N24News and Food-101 where they used pretrained models to get embeddings and two smaller ones  ROSMAP and BRCA where they trained their own encoders. They compared with over 10 baselines.

#### Results in comparison to baselines:
-  For larger datasets, in N24News  M3Col slightly outperform the baselines, while for Food-101 M3Col perform slightly worse. Overall, I would say their performance is on par with UniS-MMC

- On smaller datasets, M3Col outperforms other baselines.
#### Ablations:
The paper tested the effect of each loss they added to get the final M3Col on ROSMAP and BRCA datasets. They showed that, in fact, each loss contributes to the performance of the model in a positive way. They also validated through saliency maps that M3Col seems to focus on the correct object in comparison to other variations.

#### Calibration and Robustness:
- The paper compared M3CoL with multi-modal contrastive loss MutliSClip when predicting on random noise images and text encoder outputs. They showed that the model's confidence for M3CoL is lower in its predictions when both modalities are replaced
with random inputs in comparison to MutliSClip.
- The paper looked into the effect of corrupting one modality; they found that M3CoL is more robust to corrupt modality in comparison to MutliSClip.

**Audience:**

Yes

**Claims And Evidence:**

Yes

**Requested Changes:**

- Please give more details on how the hyperparameters are selected.
- Please add the computational cost of the training  M3CoL in comparison to other models.
- Please provide additional results showing the drop in performance when the optimal hyperparameters are not given especially for the switch between M3Co and MutliSClip.

-Typos: "Conlusions" is misspelled on page 11.

**Strengths And Weaknesses:**

## Strength

- **Clarity** -- Excellent:

  - The paper is clear and easy to follow.

- **Novelty** -- Good:

  - The approach is novel it combines mixup with multimodal and contrastive loss in a very smart way.

- **Experiments**-- Excellent:

  - The paper did a great job with the experimental section; they compared it with over 10 baselines on four datasets.

  - The ablations justify the need for different losses introduced by M3CoL.

  - The paper compared the robustness of M3CoL under conditions where one modality is random.

## Weakness

- **Method**

  - The problem with M3CoL is that it has too many losses and many hyperparameters. First, mixup $\lambda$, then the M3Co loss and the MutliSClip, and then 3 classifier losses, all of which require hyperparameters. Plus, when to switch between M3Co and MutliSClip. Reading the paper, it seems like they know the right value for all of this, which is kinda surprising, especially this ratio 0.33 M3Co + 0.67 MultiSClip. In practice, when there are many hyperparameters, it is usually very difficult to find the right ones, and as the number of losses and switches increases, complexity increases. So, training M3CoL is actually pretty difficult. Given that the performance is on par with others, it might be hard to justify why someone would use M3CoL over simpler methods.

---

> ### Author Response · Authors · 2025-03-12
> **Comment**
>
> We truly appreciate the reviewer’s recognition of this work’s strengths. The positive feedback on the clarity, readability, novelty, and experimental section means a lot. We have carefully addressed all concerns in a detailed, point-by-point response, with the changes highlighted in red in the updated manuscript.
>
>     The problem with M3CoL is that it has too many losses and many hyperparameters. First, mixup λ, then the M3Co loss and the MutliSClip, and then 3 classifier losses, all of which require hyperparameters. Plus, when to switch between M3Co and MutliSClip. Reading the paper, it seems like they know the right value for all of this, which is kinda surprising, especially this ratio 0.33 M3Co + 0.67 MultiSClip. In practice, when there are many hyperparameters, it is usually very difficult to find the right ones, and as the number of losses and switches increases, complexity increases. So, training M3CoL is actually pretty difficult. Given that the performance is on par with others, it might be hard to justify why someone would use M3CoL over simpler methods.
>
> We acknowledge that M3CoL involves multiple hyperparameters, which can increase the complexity of finding optimal hyperparameters. However, we conducted a thorough hyperparameter search on smaller datasets like ROSMAP, guided by ranges suggested in prior mixup augmentation studies (e.g., lambda and beta). The optimal ratio of 0.33 M3Co + 0.67 MultiSClip was determined empirically, supported by Liu et al., (2023). While this is a limitation, the performance gains justify the effort, especially in scenarios where capturing shared relations across modalities is critical.
> M3CoL outperforms state-of-the-art methods on three of the four datasets—N24News, ROSMAP, and BRCA (Table 2)—with competitive results on Food-101. It has been evaluated across diverse modalities, including image, text, mRNA, miRNA, and DNA, handling both two- and three-modality scenarios within a single dataset. Although performance is on par in Food-101, M3CoL shows significant improvements on low-sample datasets. Additionally, the visualization of attention heatmaps analysis (Figures 3 and 4) shows that M3CoL learns semantically meaningful multimodal representations by grounding visual concepts to relevant image regions. When tested with corrupted or random inputs, M3CoL demonstrates lower confidence scores, reducing overconfidence and enhancing reliability.
>
> Liu et al., (2023) Over-training with mixup may hurt generalization.
>
>     Please give more details on how the hyperparameters are selected.
>
> To determine the optimal hyperparameters, we conducted experiments on the ROSMAP dataset by varying three key parameters: $\alpha$, the Mixup coefficient ($\lambda$) which is sampled from a Beta distribution $\beta(\alpha, \alpha)$; $\beta$, the contrastive loss weighting factor; and the M3Co Ratio, which controls when the loss transitions from M3Co to MultiSClip. As shown in the tables below, the best-performing values were $\alpha = 0.15$, $\beta = 0.1$, and M3Co Ratio = 0.33. These values are consistent with previous findings ([1], [2]) on Mixup augmentation in unimodal settings. Any deviation from these values led to a decline in performance. We have included these results in Appendix A.2 (Hyperparameters Selection).
>
> | $\alpha$ | ACC | F1 | AUC |
> |--------|------|--------|-------|
> | 0.05 | 84.91 | 84.11 | 91.34 |
> | 0.15 | **87.74** | **87.62** | **92.98** |
> | 0.4 | 85.85 | 85.18 | 91.55 |
> | 0.8 | 83.96 | 84.11 | 91.52 |
>
> | $\beta$ | ACC | F1 | AUC |
> |--------|------|--------|-------|
> | 0.05 | 84.91 | 85.72 | 91.77 |
> | 0.1 | **87.74** | **87.62** | **92.98** |
> | 0.5 | 85.85 | 86.49 | 92.55 |
> | 1 | 85.85 | 85.72 | 92.41 |
>
> | M3Co Ratio | ACC | F1 | AUC |
> |--------|------|--------|-------|
> | 0 | 85.85 | 85.98 | 91.66 |
> | 0.2 | 85.85 | 85.71 | 92.41 |
> | 0.33 | **87.74** | **87.62** | **92.98** |
> | 0.5 | 86.79 | 86.79 | 92.58 |
> | 0.8 | 86.79 | 86.27 | 92.58 |
> | 1 | 86.79 | 86.27 | 92.58 |
>
> [1] Liu et al., (2023) Over-training with mixup may hurt generalization.
>
> [2] mixup: BEYOND EMPIRICAL RISK MINIMIZATION
>
>     Please add the computational cost of the training M3CoL in comparison to other models.
>
> We have added a discussion on the computational cost of training M3CoL in comparison to other models in Section 6. The complexity increases by a constant factor (2x) during the first one-third of training, resulting in negligible differences in training times compared to conventional contrastive learning-based multimodal frameworks.

---

> ### Author Response · Authors · 2025-03-12
> **Comment Continued**
>
> .
>
>     Please provide additional results showing the drop in performance when the optimal hyperparameters are not given especially for the switch between M3Co and MutliSClip.
>
> | M3Co Ratio | ACC | F1 | AUC |
> |--------|------|--------|-------|
> | 0 | 85.85 | 85.98 | 91.66 |
> | 0.2 | 85.85 | 85.71 | 92.41 |
> | 0.33 | **87.74** | **87.62** | **92.98** |
> | 0.5 | 86.79 | 86.79 | 92.58 |
> | 0.8 | 86.79 | 86.27 | 92.58 |
> | 1 | 86.79 | 86.27 | 92.58 |
>
> We thank the reviewer for the suggestion. As demonstrated in the attached results, the optimal performance on ROSMAP is achieved when transitioning from M3Co to MultiSClip at 1/3 of training (Acc: 87.74, F1: 87.62, AUC: 92.98). Performance drops when deviating from this optimal mixup percentage, which shows that stopping mixup augmentation after a specific phase improves classification performance. We have included these results in Appendix A.2 ((Hyperparameters Selection) to emphasize the importance of the optimal transition point.
>
>     Typos: "Conlusions" is misspelled on page 11.
>
> Thank you for catching this typo. We have corrected "Conlusions" to "Conclusions" in the revised manuscript.
>
>
> Thanks for the review once again. The review has helped refine our paper

---

> > ### Comment · Reviewer_gosT · 2025-04-16
> >
> > I want to thank the authors for the clarification and the detailed responses regarding the hyperparameters.

---

### Review · Reviewer_KokD · 2025-03-03

**Summary Of Contributions:**

The paper proposes a novel loss for training multimodal models, starting from the observations that often there does not exist a bijective correspondence between samples of the two modalities (which for examples for images and text does not hold), while typical contrastive learning approaches assume so.
 In order to deal with the many-to-many correspondence problem they propose to combine existent approach to (i) relax the objectivity assumptions with an objective by leveraging intra modal similarities (ii) use classification heads on top of each modality (and their concatenation) (iii) Apply Mixup regularization on intra-modal samples, with regularization applied only during the early training stages for optimal performance

The method is validates on 4 different benchmarks for multimodal learning, comprising the image and text modality, small scale and large scale setting and medical datasets.  Ablation studies are performed to assess to contribution of each term of the loss

**Audience:**

Yes

**Claims And Evidence:**

Yes

**Requested Changes:**

I kindly ask to address the weaknesses listed in the weaknesses section, in particular:

- As mentioned, the paper would need a better methodological contextualitazion for the proposed method  with respect to previous work  (Zou et al., 2023, Han et al 202, Gao et al., 2024, [1]).

- Better justification ( possibly experimental or via discussion) of the benefits proposed method, rather than downstream performance on benchmarks. For more details, see the weaknesses section.



_[1] Hao, Xiaoshuai, et al. "Mixgen: A new multi-modal data augmentation." Proceedings of the IEEE/CVF winter conference on applications of computer vision. 2023._

- I have some questions about the method. Some of the answers could be added to the paper to improve clarity:
     - What does the multimodal label in Cross entropy corresponds to in practice? While is easy to think about labels for each modality, is not entirely clear when this labels should be available for multiple modalities. If this is the case then, how the problem differs from  a supervised classification problem  from multiple modalities? What I the impact of the contrastive term when the labels from single modalities and shared space are representative?
     -  Does the regularization parameter $\beta$ weighting the loss term need to change across different dataset? Is there a reason behind not weighting the cross entropy terms? Some discussion and insights on this would help simplify the use of M3CoL for end users.
     - In the case multimodal labels are available and classes are perfectly separable, is it correct to say that the contrastive term of the loss is not needed?  (related to W2)
    - In the case modalities are structured differently (i.e. classes are almost perfect separable in one modality but not in the other, as it seems the case for example from the UMAP plots on News24 in Figure  7 (a) and (b) ). Does the contrastive term account for this?

**Strengths And Weaknesses:**

### Strengths

- The paper lists a large number of experiments, on different modalities and domains, including comparison with baselines, ablation studies and qualitative experiments.

- The motivations are clearly articulated: modeling intra-modal relationships and addressing the bijectivity assumption paves the way for capturing more general structural relations—an area that, to the best of my knowledge, remains underexplored in multimodal contrastive learning.

- The paper is well written and mostly easy to follow.

- The approach reaches good performance on multimodal benchmarks, obtaining SOTA in a subset of the cases.

###  Weaknesses

- The major weakness of the paper stays in the lack of a good methodological contextualization with previous work. It seem to me that the method  is in combining previous cited  work such as SoftClip ( Gao et al., 2024) to relax the bijective contrastive learning assumption, previous multimodal works using mixup (e.g. [1])   and  methods that leverage single modality classification task to improve multimodal learning ((Zou et al., 2023), (Han et al 2022)).  Differences with respect to the these work should be discussed more in depth: is the proposed approach based on combining the previous method?  If there are fundamental differences then they should be highlighted.

- The motivation behind the proposed loss could be strengthen by adding results: in particular:
    - A discussion or experimental analysis exploring whether, in some domains or tasks, only a subset of the loss terms (e.g., using a cross-entropy term instead of a contrastive term) might suffice would further strengthen the motivation behind the proposed loss. For example, Table 1 shows that the improvement over UniS-MMC is only marginal on News24 and slightly underperforms on Food 101, while Table 2 demonstrates that M3CoL performs significantly better than DYNAMICS. Does this say something about the nature of the datasets and the quality of labels available?
   - It would be nice to have some preliminary experimental evidence, or at least some discussion of the features space learned by M3CoL as the authors anticipated in the conclusive section. Some examples of interesting questions in this direction are: (i) Does the feature space adapts better/worse to novel tasks w.r.t. to  (ii) Can it be showed that some tasks that leverage the intramodal similarity structure features learned from M3CoL perform better?


- In limitations extended training times on large-scale datasets are mentioned, but no data is reported around training times.



_[1] Hao, Xiaoshuai, et al. "Mixgen: A new multi-modal data augmentation." Proceedings of the IEEE/CVF winter conference on applications of computer vision. 2023._



**minor:**

I spotted some typos:
- page 5 "parantheses" -> "parenthesis"
- section 6 "Conlusions" -> "Conclusions"

**formatting:** something looks to be off with the *Visualization of Attention Heatmaps* paragraph at page 9, probably due to negative space formatting.

---

> ### Author Response · Authors · 2025-03-12
> **Comment**
>
> We sincerely thank the reviewer for noting the strengths of our work. The positive comments on the extensive experiments, clear motivations, and well-written presentation are greatly appreciated. We have addressed the weaknesses in a point-wise manner below. The changes in the updated manuscript are highlighted in red.
>
>     The major weakness of the paper stays in the lack of a good methodological contextualization with previous work. It seem to me that the method is in combining previous cited work such as SoftClip ( Gao et al., 2024) to relax the bijective contrastive learning assumption, previous multimodal works using mixup (e.g. [1]) and methods that leverage single modality classification task to improve multimodal learning ((Zou et al., 2023), (Han et al 2022)). Differences with respect to the these work should be discussed more in depth: is the proposed approach based on combining the previous method? If there are fundamental differences then they should be highlighted.
> We thank the reviewer for the suggestion to better contextualize M3CoL with respect to prior work. Our method builds upon and extends several key ideas from recent advancements in multimodal learning:
>
> Zou et al., 2023 (UniS-MMC): M3CoL shares the use of contrastive learning for multimodal alignment but goes beyond by introducing Mixup-based augmentation to capture shared relations across modalities, which UniS-MMC does not address.
>
> Han et al., 2022 (Dynamics): While Dynamics focuses on dynamic fusion for trustworthy classification, M3CoL emphasizes learning robust representations through Mixup-based contrastive loss, enhancing generalization and interpretability.
>
> Gao et al., 2024 (SoftCLIP): SoftCLIP introduces softer cross-modal alignment, whereas M3CoL leverages Mixup to create structured noise and align mixed samples, capturing nuanced shared relations.
>
> Hao et al., 2023 (MixGen): MixGen explores multimodal data augmentation through image interpolation and text concatenation. M3CoL differs by focusing on Mixup-based contrastive learning to align mixed samples, which is more targeted at capturing shared relations rather than generating new data points.
>
> M3CoL advances multimodal learning by addressing the limitations of traditional contrastive methods and introducing a novel approach to capturing shared relations. We have introduced a dedicated subsection titled "Relation to Previous Work" in Section 5 (Related Work) to offer a more precise and comprehensive contextualization of our methodology within the existing relevant literature.
>
>
>     A discussion or experimental analysis exploring whether, in some domains or tasks, only a subset of the loss terms (e.g., using a cross-entropy term instead of a contrastive term) might suffice would further strengthen the motivation behind the proposed loss. For example, Table 1 shows that the improvement over UniS-MMC is only marginal on News24 and slightly underperforms on Food 101, while Table 2 demonstrates that M3CoL performs significantly better than DYNAMICS. Does this say something about the nature of the datasets and the quality of labels available?
>
> We thank the reviewer for the insightful suggestion. To address whether a subset of loss terms suffices, we note that UniConcat (using only cross-entropy) and other non-contrastive methods (e.g., those without alignment) underperform compared to M3CoL, particularly in low-sample settings like ROSMAP (245 samples) and BRCA (612 samples). This highlights the importance of the contrastive term, especially when training data is limited. Our proposed Mixup-based contrastive learning augments training samples by creating convex mixtures of raw data points, leading to better and more stable performance through improved contrastive representations, as supported by Lee et al. (2020).
>
> Lee et al., (2020) i-mix: A domain-agnostic strategy for contrastive representation learning.
>
> The marginal improvement on N24News (49K samples) and slight underperformance on Food-101 (60K samples) suggest that the benefits of M3CoL are more pronounced in low-sample regimes, where data augmentation and contrastive learning play a critical role. This aligns with the nature of the datasets as smaller datasets often benefit more from structured noise and shared relation learning.
> We have added this discussion in the revised manuscript in Section 4.1, emphasizing the dataset-dependent efficacy of our approach.

---

> ### Author Response · Authors · 2025-03-12
> **Comment Continued**
>
> .
>
>     It would be nice to have some preliminary experimental evidence, or at least some discussion of the features space learned by M3CoL as the authors anticipated in the conclusive section. Some examples of interesting questions in this direction are: (i) Does the feature space adapts better/worse to novel tasks w.r.t. to (ii) Can it be showed that some tasks that leverage the intramodal similarity structure features learned from M3CoL perform better?
>
> The robustness to corrupted modalities (Table 6) and lower overconfidence on random inputs (Figure 5) analyses partially demonstrate the behavior of M3CoL’s feature space in handling novel scenarios. The UMAP representations in Figure 7 (Appendix A.6) demonstrate that the multimodal feature space learned by M3CoL exhibits better class separation and robustness compared to single-modality feature spaces (e.g., image or text embeddings alone). The visualization of attention heatmaps analysis (Figures 3, 4, and 8) already demonstrates that M3CoL effectively grounds visual concepts to semantically meaningful regions, indicating its ability to capture shared relations. Additionally, the robustness to corrupted modalities (Table 6) and lower overconfidence on random inputs (Figure 5) analyses suggest that M3CoL’s feature space is more reliable and generalizable.
>
>     In limitations extended training times on large-scale datasets are mentioned, but no data is reported around training times.
>
> We acknowledge the concern about training times. Extended training times on large-scale datasets are typical in multimodal frameworks, and M3CoL exhibits similar training times compared to conventional contrastive learning-based approaches. While the complexity increases by a factor of just 2 during the first one-third of the training phase (due to Mixup-based augmentation), this does not significantly impact the overall training duration. We have added a discussion of these points in Section 6 to better contextualize the computational overhead and training times within the broader scope of multimodal learning. Thank you for the suggestion.
>
>     I spotted some typos: page 5 "parantheses" -> "parenthesis", section 6 "Conlusions" -> "Conclusions"
>
> Thanks for catching this typo. We have corrected have corrected "Conlusions" to "Conclusions" and "parantheses" to "parenthesis" in the revised manuscript
>
>     formatting: something looks to be off with the Visualization of Attention Heatmaps paragraph at page 9, probably due to negative space formatting.
>
> Thank you for pointing this out. We tried our best to solve the formatting issue in the visualization of attention heatmaps analysis paragraph on page 9 to ensure proper spacing and alignment in the revised manuscript.
>
>     Better justification ( possibly experimental or via discussion) of the benefits proposed method, rather than downstream performance on benchmarks. For more details, see the weaknesses section.
>
> We thank the reviewer for the suggestion to provide a better justification for the benefits of M3CoL beyond downstream performance. The visualization of attention heatmaps analysis (Figures 3 and 4) demonstrates that M3CoL learns semantically meaningful multimodal representations by grounding visual concepts to relevant image regions. For example, the model focuses on specific regions (e.g., "ice cream" or "falafel") when guided by text embeddings, indicating its ability to capture shared relations across modalities effectively. This highlights the interpretability and correctness of the learned feature space.
>
> Additionally, testing on random data and single-corrupt modalities (Table 6 and Figure 5) showcases M3CoL's robustness. When tested on corrupted or random inputs, M3CoL exhibits lower confidence scores compared to traditional methods, reducing overconfidence and improving reliability. Furthermore, M3CoL consistently outperforms MultiSClip in scenarios where one modality is corrupted, demonstrating its robustness to modality noise. These results validate that M3CoL not only improves performance but also enhances interpretability and robustness, making it a compelling choice for multimodal learning tasks.

---

> > ### Comment · Action_Editor_Aikd · 2025-04-08
> > **Rebuttal feedback**
> >
> > Dear Reviewer KokD,
> >
> > Thank you for your effort in reviewing this submission. It has been some days since the authors post the rebuttal, and AE also received the system prompt about the late feedback of the reviewers. Can you take the time to confirm whether the authors' rebuttal solves your concerns ASAP?
> >
> > Best,
> > AE

---

> ### Author Response · Authors · 2025-03-12
> **Comment Continued**
>
> .
>
>     What does the multimodal label in Cross entropy corresponds to in practice? While is easy to think about labels for each modality, is not entirely clear when this labels should be available for multiple modalities. If this is the case then, how the problem differs from a supervised classification problem from multiple modalities? What I the impact of the contrastive term when the labels from single modalities and shared space are representative?
>
> The multimodal label in cross-entropy corresponds to the shared ground truth label across modalities, assuming all modalities contribute to the same classification task. In practice, this label is typically available when modalities are complementary (e.g., image-text pairs for the same class). The problem differs from supervised classification by leveraging contrastive learning to capture shared relations and align modalities, even when individual modalities are imperfect or imbalanced. When labels for single modalities and the shared space are representative, the contrastive term [refer to findings (ii) in subsection 4.1] enhances robustness by enforcing consistency across modalities, improving generalization, especially in low-sample or noisy settings. This multimodal approach goes beyond simple supervised classification by integrating cross-modal relationships.
>
>     Does the regularization parameter β weighting the loss term need to change across different dataset? Is there a reason behind not weighting the cross entropy terms? Some discussion and insights on this would help simplify the use of M3CoL for end users.
>
> As noted in Appendix A.2 (Hyperparameters Selection), the loss regularization parameter $\beta$ is kept at 0.1 to keep the loss value in the same range as the other losses. The regularization parameter $\beta$ generalizes well across datasets, as it balances contrastive learning relative to fixed cross-entropy losses. Weighting cross-entropy terms separately was avoided to simplify the framework, as they are already task-specific and optimized. This design minimizes tuning complexity for end users.
>
>     In the case multimodal labels are available and classes are perfectly separable, is it correct to say that the contrastive term of the loss is not needed? (related to W2)
>
> In the hypothetical case where multimodal labels are perfectly separable and classes are perfectly distinguishable, the contrastive term might indeed become less critical, as the cross-entropy losses alone could suffice for classification. However, in real-world scenarios, perfect separability is rare, and the contrastive term helps capture shared relations and improve robustness, especially in low-sample or noisy settings. Thus, while theoretically less needed in such an ideal case, the contrastive term remains valuable for practical applications.
>
>     In the case modalities are structured differently (i.e. classes are almost perfect separable in one modality but not in the other, as it seems the case for example from the UMAP plots on News24 in Figure 7 (a) and (b) ). Does the contrastive term account for this?
>
> Yes, this is evident from the text-only and image-only performance in Tables 1 and 3. For example, on N24News, text-only models (e.g., RoBERTa) outperform image-only models (e.g., ViT), indicating better separability in text. M3CoL leverages this by aligning the weaker image modality with the stronger text modality, improving overall performance through contrastive learning, as shown in the UMAP plots (Figure 7). This demonstrates its ability to handle modality imbalance effectively.
>
> Thanks for the review once again. The review has helped refine our paper.

---

> > ### Author Response · Authors · 2025-04-24
> > **Please review our response**
> >
> > Dear Reviewer KokD,
> >
> >
> > We appreciate your valuable feedback and have provided detailed responses to all your comments. We kindly ask you to review our responses. Should you have any further questions, please do not hesitate to initiate a discussion with us.
> >
> > Thank you for your time and consideration!

---

### Decision · Action_Editor_Aikd · 2025-04-26

**Recommendation:** Accept with minor revision

**Comment:**

This submission received the comments of three reviewers. In the initial review, three reviewers mainly show concerns about lack of a good methodological contextualization, unclear motivation, too many hyperparameters and more training details that should be reported etc. After the rebuttal, the authors successfully persuade the reviewers to change the opinions. For the present, three reviewers all agree with "Acceptance" towards this submission. However, there are some remaining points raised by the reviewers, which AC thinks should be followed by the authors in the further revision.

- Hyperparameter tuning: It is not clear a priori how to select the hyperparameters for the proposed loss, nor how each term contributes to the overall objective depending on the data. While some clarifications were provided during the rebuttal, it remains unclear in which settings M3COL offers a clear advantage over simpler alternatives.

- Analysis of learned representations: A deeper analysis of the representations learned by M3COL—such as performance on downstream tasks or feature interpretability—would help clarify the method's practical utility and better contextualize its benefits.

Please following the remaining suggestion to further improve the submission, then it will reach to the acceptance without concerns from AE.

Best,

AE

**Audience:**

The topic of interest in this submission is quite related with self-supervised learning to benefit the downstream tasks. In many scenarios that are lack of sufficient annotations but full of many unannotated samples, this will be useful, especially under multi-modal area.

**Claims And Evidence:**

This paper introduces M3Col, a multimodal contrastive learning framework that addresses fundamental limitations in current approaches by combining mixup augmentation with contrastive learning across modalities. The paper is well-written and technically sound, and the extensive experiments convincingly demonstrate the effectiveness of the proposed approach.

---

> ### Author Response · Authors · 2025-05-16
> **Comment**
>
> Thank you for your suggestions.
>
> We have added a short paragraph on hyperparameter tuning in Section 3 and a brief analysis of learned representations in Section 6 in the uploaded camera-ready version.
>
>
>
> Best regards,
>
> Authors